# Blood-Based Biomarkers as Predictive and Prognostic Factors in Immunotherapy-Treated Patients with Solid Tumors—Currents and Perspectives

**DOI:** 10.3390/cancers17122001

**Published:** 2025-06-16

**Authors:** Franciszek Kaczmarek, Anna Marcinkowska-Gapińska, Joanna Bartkowiak-Wieczorek, Michał Nowak, Michał Kmiecik, Kinga Brzezińska, Mariusz Dotka, Paweł Brosz, Wojciech Firlej, Paulina Wojtyła-Buciora

**Affiliations:** 1Student Scientific Society, Poznan University of Medical Sciences, 61-701 Poznan, Poland; franciszek.kaczmarek@pm.me (F.K.); michal99.n@pm.me (M.N.); wojtek.firlej100@wp.pl (W.F.); 2Department of Biophysics, Poznan University of Medical Sciences, 61-701 Poznan, Poland; margap@ump.edu.pl; 3Physiology Department, Poznan University of Medical Sciences, 61-701 Poznan, Poland; paulinawojtyla@ump.edu.pl; 4University Clinical Hospital of Poznan, Poznan University of Medical Sciences, 61-701 Poznan, Poland; 80994@student.ump.edu.pl (K.B.); 81714@student.ump.edu.pl (M.D.); 5Faculty of Medical Sciences in Zabrze, Medical University of Silesia, 41-808 Zabrze, Poland; 81784@365.sum.edu.pl; 6Department of Social Medicine and Public Health, Calisia University, 62-800 Kalisz, Poland

**Keywords:** immunotherapy, rheology, biomarkers, cancer, CAR, TCR

## Abstract

Immunotherapy has transformed solid tumor treatment, but cost-effective blood biomarkers for predicting and monitoring response are still needed. Easily measured tests, such as cell counts (monocytes, MDSCs, Tregs, and eosinophils), cytokines (IL-6, IL-8, and IL-10), LDH, CRP, soluble checkpoints, and ctDNA, alongside simple ratios (NLR, LMR, and PLR) and blood viscosity metrics, provide non-invasive prognostic and predictive insights. Combining these biochemical and rheological markers could improve patient selection and optimize immunotherapy outcomes.

## 1. Introduction

Immunotherapy has transformed cancer treatment by heralding a new era in oncology. By targeting key signaling pathways—PD-1/PD-L1, CTLA-4/CD28, JAK/STAT, PI3K/AKT/mTOR, and MAPK/ERK—it enhances T-cell activation and proliferation and regulates immune checkpoint expression, such as PD-L1 on tumor cells. Dysregulation of these pathways can result in immunosuppression and therapeutic resistance, making them critical targets for intervention. Current approaches include immune checkpoint inhibitors, adoptive cell transfer, cancer vaccines, and cytokine therapies, all designed to bolster the immune system’s capacity to recognize and eradicate malignant cells. Nevertheless, tumor-intrinsic resistance—driven by genetic alterations, metabolic rewiring, and influences from the tumor microenvironment—remains a significant challenge to achieving durable responses [1,2].

Immunotherapy includes, among others, immune checkpoint inhibitors (ICIs), cancer vaccines, monoclonal antibodies, and adoptive cell therapy. A significant part of cancer treatment is the ability to determine such markers to help optimize treatment. This work focuses on cost-effective blood biomarkers that can be used as prognostic and predictive factors in immunotherapy-treated patients with solid tumors.

The current Food and Drug Administration-approved predictive markers in patients treated with ICIs are tumor mutational burden (TMB), programmed cell death ligand 1 (PDL1) expression, and microsatellite status (MSI). A commonly used blood-based biomarker is ctDNA; however, only in combination with other complementary biomarkers, such as immunophenotyping of mononuclear cells, extracellular vesicle analysis, and plasma proteomics, can it provide sufficient information to determine whether patients will benefit from ICIs [2,3,4].

ICIs have transformed cancer treatment, yet only a fraction of patients experience durable benefits, highlighting the need for robust predictive biomarkers that integrate both tumor-intrinsic and host-immune factors. Peripheral blood cell ratios—such as the neutrophil-to-lymphocyte ratio (NLR), lymphocyte-to-monocyte ratio (LMR), and platelet-to-lymphocyte ratio (PLR)—have gained prominence because they reflect the balance between tumor-promoting inflammation and anti-tumor immune competence, are readily derived from routine complete blood counts, and can be monitored over time to capture changes in systemic immunity [1]. Equally critical are tumor-intrinsic biomarkers such as PD-L1 expression, which directly measures engagement of the PD-1/PD-L1 axis in the tumor microenvironment and remains the most widely implemented assay for selecting patients for anti-PD-1/PD-L1 therapies [2,3]. In parallel, ctDNA provides a minimally invasive, dynamic readout of tumor burden and early treatment effect, with rapid declines in ctDNA levels after ICI initiation correlating with superior progression-free and overall survival (OS) independently of PD-L1 status or TMB [1,2,3,4]. More exploratory approaches, including rheological parameters that assess tumor-induced alterations in blood flow and viscosity, aim to capture the impact of systemic inflammation on immune-cell trafficking and endothelial interactions. By combining tumor-intrinsic markers, systemic inflammation indices, and real-time circulating measures, a multi-dimensional biomarker framework can more precisely stratify patients and guide personalized immunotherapy strategies.

Hemorheological changes—such as alterations in whole blood viscosity, plasma viscosity, and erythrocyte aggregability—are observed in a wide range of diseases [1,2,3]. The key challenge, therefore, is to skillfully distinguish those changes that are specific to the tumor–immunotherapy interaction from systemic alterations driven by general inflammation or cancer progression, regardless of treatment efficacy. Hemorheological parameters reflect the complex interplay of tumor-induced inflammation, systemic disease effects, and therapy-related changes; thus, isolating immunotherapy-specific signals requires careful analysis [2,3,4]. Longitudinal studies tracking these parameters before, during, and after immune checkpoint inhibitor therapy are essential, as they can reveal temporal patterns that differentiate treatment effects from changes due to disease progression or inflammatory processes. Integrating hemorheological data with established biomarkers—such as the neutrophil-to-lymphocyte ratio, CRP levels, or circulating tumor DNA—will further facilitate the distinction between immunotherapy-related alterations and systemic responses, creating a more reliable predictive framework [1,2,3]. The application of advanced analytical methods and multivariate statistical models can help identify characteristic hemorheological signatures associated with a positive response to immunotherapy. Finally, the development and adoption of standardized protocols for measuring and interpreting hemorheological parameters in the context of immunotherapy are crucial to confirm their specificity and ensure reproducibility. Together, these strategies will strengthen the role of hemorheological parameters as cost-effective, tumor–immunotherapy–specific biomarkers, distinct from general markers of inflammation or disease progression.

In our review, we focused on critical pathways involved in cancer development, systemic inflammatory response markers (NLR, LMR, and PLR), and rheological parameters as prognostic and predictive factors in immunotherapy.

## 2. Key Signaling Pathways in Cancer Immunotherapy

The main molecular mechanisms involved in immunotherapy and cancer resistance strategies include numerous signaling pathways and immune checkpoint routes. One example is the PD-1/PD-L1 pathway, in which the programmed cell death protein 1 (PD-1) on T lymphocytes binds to its ligand PD-L1 on tumor cells, leading to T cell exhaustion and reduced anti-tumor activity. PD-1/PD-L1 inhibitors, such as nivolumab and pembrolizumab, block this interaction, restoring T cell capability to attack cancer cells. Similarly, the CTLA-4 protein competes with CD28 for binding to CD80/CD86 on antigen-presenting cells, inhibiting T cell activation. A CTLA-4 blockade using ipilimumab enhances T cell activation and proliferation, strengthening the immune response [5] (Figure 1).

Cytokine signaling, in turn, involves the activity of interferon-gamma (IFN-γ), which activates the JAK/STAT pathway, leading to increased expression of MHC molecules and PD-L1 on tumor cells. Although this may enhance antigen presentation, it simultaneously provides inhibitory signals through PD-L1. Mutations in JAK1/2 or STAT1 can lead to resistance to immunotherapy. Interleukin-2 (IL-2) promotes T cell proliferation and activation via the PI3K/Akt/mTOR pathway; however, high doses of IL-2 may also stimulate regulatory T cells (Tregs), suppressing the immune response [6] (Figure 1).

Innate immunity pathways, on the other hand, involve mechanisms such as the activity of cyclic GMP-AMP synthase (cGAS), which detects cytosolic DNA and produces cGAMP that activates the STING protein, leading to the production of type I interferons via TBK1 and IRF3. Activation of this pathway enhances dendritic cell maturation and T cell priming. Toll-like receptors (TLRs) recognize pathogen-associated molecular patterns, activating the NF-κB, MAPK, and IRF3/7 pathways, which induce the production of pro-inflammatory cytokines and type I interferons. TLR agonists are used as adjuvants in cancer vaccines to enhance the immune response [7] (Figure 1).

Immunotherapy also leverages adaptive immune cell signaling mechanisms. Engagement of the T cell receptor (TCR) with antigen-MHC complexes, along with co-stimulation (e.g., CD28), activates the PI3K/Akt/mTOR pathway, promoting T cell survival, proliferation, and effector functions. Second-generation CAR-T cells incorporate co-stimulatory domains such as CD28 or 4-1BB to enhance T cell activation and persistence [8] (Figure 1).

Despite the efficacy of immunotherapy, cancer cells may develop resistance through various mechanisms. Loss-of-function mutations in JAK1/2 impair IFN-γ signaling, reducing MHC-I and PD-L1 expression, which leads to resistance to PD-1 blockade. PTEN loss activates the PI3K pathway, promoting an immunosuppressive tumor microenvironment and resistance to immunotherapy. Tumor cells inhibit the PI3K/Akt/mTOR pathway in T cells, impairing their function. Adenosine, produced in the tumor microenvironment, activates A2A receptors on T cells, suppressing their activation and promoting immunosuppression. Low oxygen levels in tumors upregulate HIF-1α, which can promote immunosuppressive pathways and reduce immune cell infiltration. Myeloid-derived suppressor cells (MDSCs) and regulatory T cells (Tregs) suppress immune responses through various mechanisms, including ARG1, iNOS, and TGF-β signaling [9] (Figure 1).

## 3. Blood Based Biomarkers

### 3.1. Inflammatory Response Markers—NLR, LMR, and PLR

#### 3.1.1. NLR

The neutrophil-to-lymphocyte ratio (NLR) is a biomarker derived from the proportion of neutrophils to lymphocytes in peripheral blood. It reflects the dual nature of the immune system, combining the innate immune response—primarily driven by neutrophils—with the adaptive immune response, mediated by lymphocytes. Neutrophils serve as the body’s first line of defense against invading pathogens, utilizing various mechanisms such as chemotaxis, phagocytosis, the release of reactive oxygen species (ROS), and granular proteins, as well as the production and secretion of cytokines [10].

NLR serves as a biomarker reflecting both chronic inflammation and immune function. An increased NLR has been linked to cardiovascular conditions, autoimmune disorders, sepsis, cancer, and higher overall mortality rates in the general population [11,12]. In patients with malignant solid tumors, NLR may reflect the inflammatory response triggered by neoplastic cells. High NLR was associated with increased peritumoral macrophage infiltration, high levels of pro-inflammatory cytokines, and a high number of tumor-associated neutrophils (TANs) [12]. An umbrella review by Cupp MA et al. analyzed 204 meta-analyses from 86 studies investigating the relationship between NLR or TANs and cancer outcomes. Strong evidence linked NLR to outcomes in composite cancer endpoints, cancers treated with immunotherapy, and some site-specific cancers (urinary, nasopharyngeal, gastric, breast, endometrial, soft tissue sarcoma, and hepatocellular cancers) [13].

A comprehensive meta-analysis by Wang H. et al. showed that high NLR before treatment ICIs for gastric tumors (GCs), hepatocellular carcinomas (HCCs), head and neck squamous cell carcinomas (HNSCCs), melanomas, non-small-cell lung cancers (NSCLCs) and renal cell carcinomas (RCCs) had been observed with worse OS and progression-free survival (PFS) outcomes. Moreover, high pretreatment NLR may be associated with a lower objective response rate (ORR) for GC, HCC, HNSCC, and NLSC and a disease control rate (DCR) for HCC and HNSCC. A sub-analysis of the INVIDIa-2 study on influenza vaccination in cancer patients (including NSCLC, RCC, and melanoma) treated with ICIs found NLR below 3, 4 to be an independent prognostic factor for OS [14,15]. Low baseline NLR values may also increase the risk of developing irAE in patients with NSCLC, which may be related to the effectiveness of treatment [16]. However, high post-treatment NLR was identified as a potential predictive biomarker for the occurrence of irAEs in patients with different solid tumors treated with ICIs [17]. High NLR during ICIs treatment was associated with significantly worsened OS and PFS rates [18,19,20,21,22], disease control, and treatment response [20]. Guo Y et al. suggested that NLR levels in patients who respond to immunotherapy remain stable but may increase in those who do not respond to treatment. More importantly, the dynamic changes in NLR levels following ICI therapy significantly impact OS, PFS, and ORR rates [21]. A related indicator is the derived neutrophil–lymphocyte ratio (dNLR), calculated using the formula dNLR = ANC/(WBC − ANC). Elevated dNLR levels are associated with poorer OS and PFS [19,22].

NLR has been combined with other markers to more accurately predict the response to ICIs. Most notable of these risk stratification scores are the Lung Immune Prognostic Index (LIPI) and the Lung Immuno-oncology Prognostic Score (LIPS-3), both for advanced NSCLC patients. LIPI uses dNLR with serum LDH level [23], while LIPS-3 combines NLR, ECOG performance score, and pretreatment steroid status [24]. High dNLR and LDH were correlated with shorter PFS and OS in other types of solid tumors treated with ICIs (gastrointestinal, breast, and gynecological) [25].

#### 3.1.2. LMR

The lymphocyte-to-monocyte ratio (LMR) is a parameter determined by dividing the number of lymphocytes by the number of monocytes in a blood test and is used to analyze inflammation and the body’s immune response. Lymphocytes play a key role in anti-tumor defense, while monocytes can support tumor development by producing pro-inflammatory factors. A low LMR may indicate weakened immunity and is associated with poorer prognosis in cancer patients. In contrast, a higher LMR correlates with better treatment outcomes and is being studied as a potential prognostic marker [26].

LMR is used as an immunotherapy efficiency biomarker for the treatment of gastric cancer patients. The high value of LMR is associated with a better overall prognosis, contrary to high values of NLR and PLR [27]. This may be caused by the immunological role of lymphocytes, as they act as anti-tumor cells. For instance, they mediate anti-tumor response. Similar combinations of those parameters’ values were found among melanoma and gastric patients [19]. A low LMR is linked to an unfavorable prognosis in lung cancer, colorectal cancer (CRC), renal cell carcinoma, and melanoma [28].

#### 3.1.3. PLR

The platelet-to-lymphocyte ratio (PLR) is a novel hematological biomarker that has gained attention for its potential role in assessing systemic inflammation and prognosticating outcomes in various pathological conditions. Emerging evidence indicates that an elevated PLR is strongly correlated with heightened inflammatory activity, the pathogenesis of atherosclerosis, and enhanced platelet reactivity, suggesting its relevance as a predictive indicator in clinical settings [29]. Thrombocytosis, characterized by a platelet count (PLT) exceeding 450,000/µL, is commonly observed in patients with solid tumors and chronic inflammatory conditions. Its coexistence with enhanced platelet activation can substantially increase the predisposition to thrombotic complications. PLR has been recognized as a potential prognostic biomarker in malignancies such as ovarian, breast, and lung cancer. However, its prognostic relevance in CRC remains a subject of ongoing investigation and lacks definitive consensus [30].

A high PLR level before treatment is associated with worse OS and PFS outcomes in patients with advanced cancer, particularly in metastatic renal cancer, where the worst outcomes were observed [31]. PLR is an effective indicator of the risk of cancer recurrence after liver transplantation [32].

NLR, PLR, and MLR are cost-effective, easily obtainable, and reliable indicators of systemic inflammation and are derived from white blood cell counts. These inflammatory ratios may offer greater predictive accuracy in assessing inflammation compared to individual measurements of neutrophils, platelets, monocytes, or lymphocytes, as they are less influenced by confounding factors. Research has shown that they can serve as biomarkers of inflammation and indicators of poor prognosis in various diseases, including cancer [33] (Figure 2).

### 3.2. Rheological Parameters

Mechanics is the field of science that explores how material bodies move and deform when subjected to external forces. Within this discipline, rheology, often called the study of flow, focuses specifically on how different materials respond to mechanical forces, particularly in terms of their ability to deform and flow [34]. Hemorheology combines two distinct aspects: its primary objective lies in medicine, whereas its research approaches are derived from science and technology. Consequently, conducting advanced studies in this field necessitates collaboration between specialists from both areas [35]. Blood rheology plays a crucial role in regulating tissue perfusion, and based on the Poiseuille relation, hemodynamic resistance in a vascular network with constant geometry is directly proportional to blood viscosity. Additionally, blood rheology can affect vascular tone by altering wall shear stress, which influences the endothelial production of vasoactive substances such as nitric oxide [36].

Important parameters influencing the rheological properties of blood are hematocrit, plasma viscosity, and the aggregability and deformability of erythrocytes. Plasma viscosity, in particular, is a forgotten parameter [37]. Plasma viscosity is mainly determined by the presence of high-molecular proteins such as fibrinogen, immunoglobulins, and lipoproteins [38,39]. Excessive plasma viscosity may also result from both monoclonal and polyclonal disorders associated with an increase in immunoglobulins [40]. Figure 3 presents the determinants of whole blood viscosity. It is also worth noting that the analysis of changes in the values of hemorheological parameters, such as aggregability of erythrocytes and plasma viscosity, may be significant because thromboembolic events may be the cause of complications in the process of surgical and chemotherapeutic treatments of cancer.

Changes in the values of hemorheological parameters are also related to such aspects of RBC pathophysiology as reduced deformability, increased adhesion, and changes in the properties of the erythrocyte membrane. This has been observed in association with microcirculation dysfunction in many diseases, including hereditary hemoglobinopathies, hemolytic anemias, malaria, sepsis, lupus, blood transfusion complications, kidney disease, cancer, diabetes, obesity, cardiovascular disease, neurological disorders, and heavy metal exposure [41,42,43].

Regarding the physical properties of blood, rheology can be widely used to assess blood parameters. It is especially crucial for previously described hematological malignancies where basically the blood is the most affected tissue. Hematological malignancies are often unveiled by analyzing the rheological parameters of blood samples collected from patients. Immunotherapy, i.e., CART-T cell therapy or kinase inhibitors, is one of the major parts of the treatment of these malignancies. Major studies concluded that whole blood viscosity (WBV) is increased in myeloproliferative neoplasms [44,45,46]. This can lead to a higher risk of thrombotic complications, such as deep vein thrombosis [47]. It is reported that COVID-19 vaccinations among patients with multiple myeloma, which is described to cause high WBV by itself, are prone to cause adverse effects, including thrombosis. Hematocrit value, which does not require advanced rheometer to be assessed, is proposed as a parameter to measure the efficiency of T-cell harvesting of CD3+ cells for T-cell therapy regarding hematological neoplasms [48]. Additionally, in mice models the efficiency of immunotherapy can be measured by Doppler ultrasonography [49]. Increased perfusion in tumors’ vessels contributes to more efficient immune checkpoint blockade (ICB).

However, the data for solid tumors affecting rheological blood properties is firmly limited. Some studies aim to establish rheological parameters for patients with solid tumors treated with immunotherapy agents. For instance, tumor microenvironment (TME) is closely associated with the blood flow within tumor mass. The most critical rheological parameter, i.e., whole blood viscosity, is increased in vessels covering TME. It is reported that it is prone to tumor metastasis and has a worse response during the treatment [50]. In newly diagnosed cancer cases, a common pattern involves elevated plasma viscosity (PV) accompanied by increased red blood cell (RBC) aggregation. This results in a state of hyperviscosity, which is usually mitigated by anemia. However, if hematocrit levels rise uncontrollably in cancer patients, it can impair microcirculatory blood flow, potentially leading to a harmful hemorheological condition [51].

The number of articles related to solid tumors is limited; however, Han JW et al. demonstrated that nivolumab-treated patients with HCC who had lower WBV tended to have better OS and PFS rates. However, these results did not reach statistical significance (OS and PFS rates: *p* = 0.069 and *p* = 0.067, respectively), and the study group was small (*n* = 33). The high diastolic whole blood viscosity (WBV) (>16.0 cP) was associated with poorer OS and PFS compared to the group with lower WBV (≤16.0 cP). The differences were close to statistical significance (*p* = 0.069 for OS and *p* = 0.067 for PFS, which may be due to the small sample size). The objective response rate was 25.0% in the low WBV group (6/24, including one complete response and five partial responses) compared to 0% in the high WBV group (0/9), although the difference was not statistically significant (*p* = 0.409). The study suggests that WBV may serve as a potential biomarker in HCC, particularly in the context of immunotherapy, but emphasizes the need for larger, prospective studies [52].

In a retrospective study evaluating 100 oncology patients treated with ICIs, focusing on sarcopenia as an independent predictor of responses to immunotherapy, Ucgul E et al. demonstrated that elevated sedimentation rates could serve as a significant prognostic indicator for OS rate [53]. Issa M et al. showed that low hemoglobin levels in patients with HNSCC treated with ICIs are an independent predictor of poor OS rate [27]. Higher baseline hemoglobin (above 11 g/dL) was associated with better PFS and OS in NSCLC patients treated with ICIs [40]. Similar conclusions came from a subsequent study, which showed that higher hemoglobin, along with higher Treg lymphocytes, MPV (mean platelet volume), and lower monocytes were predictive factors for PFS and OS in this cancer [54,55]. The results of aforementioned studies seem to align with the findings of He Y et al., who demonstrated that baseline hemoglobin levels are correlated with OS, PFS, and ORR in patients treated with ICIs. Furthermore, the correlation between hemoglobin levels and treatment results with ICIs was independent of the treatment approach (chemotherapy before ICIs, ICI monotherapy, or ICI combination therapy) and other factors such as age, gender, cancer stage, or TMB. Additionally, it was demonstrated that combining TMB assessment with hemoglobin levels can increase the predictive accuracy of treatment response to ICIs [56]. A red cell-based score was developed by Mazzaschi et al. for metastatic renal cell carcinoma (mRCC) patients. Hemoglobin ≥ 12 g/dL, MCV > 87 fL, and RDW ≤ 16% are considered as positive factors. Initially validated in a group of patients treated with TKI, it has proved to be prognostic in patients treated with first-line immunotherapy (TKI plus ICI or ICI plus ICI), as well as second-line nivolumab [57,58,59].

Table 1 presents the available data on the impact of blood rheological parameters on immunotherapy outcomes across various cancer types. In some cases, although certain trends were observed, the results did not reach statistical significance, which may be attributed to the small sample sizes of the study populations.

Data on the impact of solid tumors on blood rheological properties remain limited; however, immunotherapy—through its effects on the tumor microenvironment and inflammatory processes—offers plausible mechanisms by which hemorheological parameters could serve as predictors of treatment response. The referenced study in HCC, along with observed associations with inflammatory markers such as NLR and PLR, supports this hypothesis. Further research, particularly in larger patient cohorts and using standardized methodologies, is essential to confirm these mechanisms and establish their clinical utility.

### 3.3. Integrated Significance of Inflammatory Markers and Blood Rheological Parameters in Assessing the Response to Immunotherapy

In recent years, there has been growing interest in the use of hematological inflammatory markers and blood rheological parameters as readily accessible and non-invasive biomarkers for monitoring the effectiveness of immunotherapy in oncology patients. The most commonly analyzed indicators include NLR, LMR, and PLR, which reflect the balance between the body’s inflammatory and immune responses.

Elevated NLR and PLR—indicating a predominance of the inflammatory component over the lymphocytic one—have been significantly associated with poorer prognosis and lower treatment efficacy in patients with various solid tumors, including non-small-cell lung cancer (NSCLC), HCC, renal cell carcinoma (RCC), and melanoma [11,12,13,14,15,16,17,18,19,20,21,22,29,30,31,32]. In contrast, a high LMR, reflecting a stronger lymphocytic response over the protumorigenic activity of monocytes, correlates with better treatment outcomes and longer OS [26,27,28]. Moreover, not only baseline values but also dynamic changes during therapy may provide valuable prognostic and predictive insights, highlighting the potential of tracking these ratios throughout immunotherapy [21].

Complementing these markers are blood rheological parameters, which describe the physical properties of blood flow, such as whole blood viscosity (WBV), plasma viscosity (PV), hematocrit, erythrocyte deformability, and red blood cell aggregation. Changes in these parameters, especially in the context of tumor microcirculation, may influence drug delivery effectiveness and the immune response [36,37,38,39,40]. Elevated WBV can impair tissue perfusion and reduce the efficacy of ICIs, as observed, for instance, in HCC patients treated with nivolumab [52]. Although data regarding solid tumors remain limited, current studies suggest the prognostic potential of rheological parameters, particularly when combined with classical hematological indices.

It is also worth noting parameters such as hemoglobin levels, hematocrit, and red blood cell volume and distribution indices (MCV and RDW), which have been identified as independent prognostic factors in the context of immunotherapy in patients with NSCLC, HNSCC, and renal cell carcinoma [27,40,54,55,56,57,58,59]. High hemoglobin levels correlate with improved PFS and OS regardless of the type of immunotherapy used, and combining hemoglobin with TMB assessments enhances the predictive accuracy for treatment response [56].

In summary, both inflammatory markers (NLR, LMR, and PLR) and blood rheological parameters provide valuable information regarding inflammation status, immune function, and circulatory efficiency in patients undergoing immunotherapy. Their combined use may offer a foundation for more personalized treatment approaches and more accurate predictions of therapeutic efficacy, although further research is needed to validate their clinical utility [33,50,51,52].

### 3.4. Blood-Based Biomarkers in Preclinical and Clinical Studies

In cancer immunotherapy, preclinical mouse models, especially humanized ones with human immune cells, are essential for testing new treatments. These models enable researchers to measure blood-based biomarkers, such as cytokine levels and immune cell ratios, to assess the efficacy and safety of immunotherapy. Studies have shown that changes in these biomarkers can indicate treatment success or potential side effects, providing valuable insights for patient care [60,61].

Our research focuses on preclinical and clinical studies that describe less commonly investigated parameters, which may be considered in future efforts to develop biomarkers of immunotherapy responses (Table 2 and Table 3). However, more evidence is emerging from clinical studies involving patients, as these are more valuable and provide a better understanding of these parameters in vivo [62,63,64].

In preclinical studies using various animal models (Table 2), peripheral blood biomarkers—particularly neutrophil counts and phenotypes—were found to strongly correlate with immunotherapy efficacy [65,66]. First, therapy-elicited neutrophils bearing an interferon gene signature proved essential for successful anti-tumor responses [65]. Second, host genetics and tumor type determined both the proportion of circulating neutrophils and their capacity to control tumor growth [66]. Moreover, neutrophil activation (e.g., via TNF, a CD40 agonist, or tumor-targeting antibodies) induced tumor eradication through oxidative damage [67]. In humanized models, improved reconstitution of human neutrophils enables the calculation of NLR as a translational marker [68], and measurement of cytokine release (e.g., IL-6) facilitates assessment of cytokine release syndrome risk [69]. Finally, in HIS-BRGS mice, the degree of human chimerism and distribution of T-cell subsets in circulation correlated with tumor infiltration, underscoring the value of liquid-biopsy immune profiling for therapy monitoring [70].

Table 3 summarizes key original studies investigating blood-based biomarkers in immunotherapy for solid tumors. In non-small-cell lung cancer (NSCLC), early work showed that higher circulating CD4/CD8 ratios and lymphocyte percentages correlate with improved responses to anti-PD-(L)1 therapy, whereas elevated PD-1+ T-cell frequencies, neutrophil-to-lymphocyte ratio (NLR), and monocyte-to-lymphocyte ratio (MLR) predict poorer outcomes [71]. Several studies have focused on blood TMP (bTMB) in NSCLC treated with atezolizumab or other ICIs: one study demonstrated that bTMB reliably identifies patients with significant PFS benefit, though optimal cutoffs and assay refinements remain under investigation [72,73], while subsequent work has shown that adjusting bTMB for ctDNA or maximum somatic allele frequency (Ma-bTMB and LAF-bTMB) improves its predictive power for both OS and PFS [74,75,76]. Another NSCLC study found that detection of circulating tumor cells alongside ctDNA-adjusted bTMB identifies patients less likely to achieve durable responses [77]. In melanoma, on-treatment increases in exosomal PD-L1 levels distinguished pembrolizumab responders from nonresponders [78], and allele frequency-adjusted exosomal markers such as PD-1 and CD28 on T cells were associated with longer PFS and OS following ipilimumab [79]. Together, these findings highlight the evolving landscape of blood-based biomarkers—from cellular ratios to ctDNA-based measures—in predicting immunotherapy efficacy across tumor types (Table 3).

### 3.5. New Players in Immunotherapy: Molecular Basis of Emerging Peripheral Blood Biomarkers

In the search for predictive molecular markers of responses to immunotherapy, particular attention is paid to inflammatory pathways activated in neutrophils and their interactions with the tumor microenvironment. Activation of the STAT3 pathway in neutrophils leads to the overexpression of PD-L1, which inhibits the function of cytotoxic T lymphocytes and weakens the anti-tumor response. Blocking both STAT3 and PD-L1 could improve the effectiveness of immunotherapy. Additionally, circular RNAs, such as CircPACRGL, regulate neutrophil functions by increasing TGF-β levels, promoting the differentiation of neutrophils into the N2 phenotype with immunosuppressive properties. The presence of this circRNA and activation of the TGF-β pathway may serve as biomarkers predicting the lack of response to immunotherapy and offer potential directions for combination therapies targeting the inflammatory components of the tumor microenvironment [80].

Platelets play a significant role in tumor progression, particularly by enhancing angiogenesis through VEGF, the levels of which are elevated in cancer patients. Pro-inflammatory cytokines such as IL-1 and IL-6 stimulate increased megakaryocyte production, leading to thrombocytosis, which is considered a negative prognostic factor. Additionally, within the tumor microenvironment, platelets become activated and release factors that support neovascularization and tumor growth. As a result, in the context of chronic inflammation and immunosuppression, a decrease in lymphocyte count occurs, contributing to an elevated PLR, which has been associated with poorer prognosis in patients with solid tumors [81].

Cancer cells can contribute to lymphopenia in several ways. One of them is by directly inducing the death of lymphocytes through the expression of pro-apoptotic ligands. Additionally, tumors can weaken the ability of T lymphocytes to respond to stimuli by disrupting the signaling of their receptors. An important role is also played by the increased number of regulatory T cells, which, through the expression of the CTLA-4 molecule, promote immunosuppression and can lead to the death of other immune cells in an activation-dependent manner [27]. Lymphocytes support the survival of CRC patients by inducing tumor cell apoptosis in the immune response. In contrast, monocyte activation inhibits the proliferation and activation of T cells, leading to immunosuppression and weakened anti-tumor responses. A high number of monocytes in peripheral blood is associated with worse prognosis in cancer patients. Monocyte activation inhibits the proliferation and activation of CD4+ CD25- and CD8+ CD25-T cells, resulting in immune suppression [26].

### 3.6. ctDNA

Circulating tumor DNA (ctDNA), a type of cell-free DNA shed into the bloodstream by tumor cells, reflects tumor burden and offers a real-time view of tumor dynamics [82] (Figure 4). Beyond its diagnostic potential, ctDNA can stimulate innate immune pathways, promoting systemic inflammation and immune dysregulation. Through recognition by pattern recognition receptors (PRRs) such as Toll-like receptors (TLRs) and cytosolic DNA sensors, ctDNA activates inflammatory signaling cascades, including the caspase-1 inflammasome pathway, leading to the release of pro-inflammatory cytokines like IL-1β and IL-18 [82,83].

ctDNA is a valuable tool for monitoring tumor burden and tumor dynamics in real time. However, ctDNA, although reflecting genomic alterations, may not always fully capture the complex spatial heterogeneity (differences between tumor regions or metastases) or clonal evolution (emergence of new mutations or subclones over time) of the tumor [84]. ctDNA levels may be low in early stages of disease or in low-burden tumors, which limits its sensitivity. Furthermore, the ctDNA fraction may be variable and may not always reflect all tumor clones, especially those with low prevalence. To overcome these limitations, it is necessary to combine ctDNA analyses with other methods, such as mononuclear cell immunophenotyping, extracellular vesicle analysis, and plasma proteomics, as well as integrated approaches with circulating tumor cells (CTCs) [85,86].

Although ctDNA provides valuable real-time information on tumor dynamics, its limitations in fully reflecting tumor heterogeneity and clonal evolution emphasize the need for multimodal approaches. Combining ctDNA with other biomarkers and advanced analytical methods may provide a more comprehensive view of tumor biology and treatment response [87].

This inflammatory response is part of a broader tumor-induced immune disruption, which also includes enhanced lymphangiogenesis and lymphatic remodeling, driven by VEGFC and VEGFD, facilitating tumor dissemination. Tracking ctDNA levels offers a non-invasive biomarker for monitoring tumor progression and treatment response. A decline in ctDNA is typically associated with therapeutic success, while increasing levels may signal resistance or relapse. Its dual role as a tumor burden marker and inflammatory trigger makes ctDNA a valuable tool in both cancer diagnostics and immunological research [88].

### 3.7. LDH and CRP

Lactate dehydrogenase (LDH) and CRP are widely used markers of systemic inflammation that are frequently elevated in cancer (Figure 4). These acute phase reactants reflect tumor-associated tissue damage, immune activation, and overall inflammatory burden, often correlating with advanced disease and poor prognosis [89,90].

LDH, a metabolic enzyme released during cell damage, is linked to increased tumor burden, rapid cell turnover, and aggressive tumor phenotypes. It also contributes to tumor progression by promoting angiogenesis and immune evasion. Ratios such as the LDH-to-albumin ratio (LAR) have shown a prognostic value in CRC and correlate with TNM staging [89].

CRP, produced by the liver in response to pro-inflammatory cytokines like IL-6, is a sensitive marker of systemic inflammation. Elevated CRP levels have been associated with tumor growth, metastasis, and therapy resistance in multiple malignancies [91]. Its broad availability and responsiveness to inflammation make it a practical tool for monitoring cancer progression and treatment response.

For LDH and CRP, ROC-AUC analysis in melanoma patients showed that CRP (AUC = 0.933) was superior to LDH (AUC = 0.491) in discriminating stage IV melanoma. In the context of diagnosing stage IV melanoma, at a cutoff of 3.0 mg/l, CRP achieved a sensitivity of 0.769 and a specificity of 0.904, whereas LDH did not provide additional information compared with CRP [92].

In NSCLC, a study showed that both serum CRP and LDH levels, as well as their combination, could predict the response to checkpoint inhibitors. High LDH levels and low CRP levels were associated with unfavorable progression-free survival [93]. In acute pancreatitis, LDH showed higher sensitivity (94.9%) but lower specificity (88.2%) compared to CRP (sensitivity of 59.0% and specificity of 97.4%) [94].

In lung cancer, although the combination of CRP and neuron-specific enolase (NSE) showed high specificity (94%) and accuracy (82.67%), LDH levels were not statistically different in the patient groups [95].

Analysis of the results obtained from the studies shows that CRP appears to be superior to LDH in staging melanoma, whereas their individual diagnostic powers differ in other conditions, such as acute pancreatitis. Importantly, studies in NSCLC and prostate cancer show that their combination can increase predictive power. This suggests that rather than one marker being universally “better,” they may offer complementary information, with their optimal utility being dictated by the specific tumor type, stage, and clinical question.

### 3.8. Cytokine Signaling

Cytokines are key signaling molecules that regulate immune responses and inflammation within the tumor microenvironment (TME). Their effects vary depending on the tumor type, stage, and host immune status, and they significantly influence tumor progression, immune evasion, angiogenesis, and metastasis [96]. Among them, interleukins like IL-6, IL-8, and IL-10 play prominent roles in shaping tumor behavior and immune cell activity [96,97,98] (Figure 4). Due to their broad impact on immune signaling, cytokines are being actively studied as biomarkers and therapeutic targets in cancer [96].

IL-6 and IL-8 are strongly associated with pro-tumorigenic activities. IL-6 promotes tumor cell proliferation, survival, and metastasis via the JAK-STAT3 pathway, which upregulates genes involved in cell cycle progression and angiogenesis [96,97]. It also suppresses anti-tumor immunity by promoting the expansion of myeloid-derived suppressor cells (MDSCs) and inhibiting cytotoxic T lymphocytes (CTLs) [96]. Elevated IL-6 is correlated with advanced stages in cancers like breast and bladder cancer [96,97]. IL-8, a chemokine, supports tumor progression by promoting angiogenesis, tumor invasion, and extracellular matrix degradation via matrix metalloproteinases (MMPs). It also attracts neutrophils, sustaining chronic inflammation within the TME [96,98].

IL-10 has a dual role in cancer. Known for its anti-inflammatory effects, IL-10 suppresses pro-inflammatory cytokine production and enhances regulatory T cell (Treg) development, thereby dampening anti-tumor immune responses. While this may help prevent tissue damage from chronic inflammation, IL-10 can also facilitate tumor growth by suppressing immune surveillance. Its role is highly context-dependent, making therapeutic targeting of IL-10 complex but potentially valuable in modulating the TME [96].

Studies show that dynamic changes in cytokine profiles, such as decreased IL-6 and IL-8 levels, may reflect the short-term efficacy of immunotherapy [99]. Changes in CCL11, IL1RA, and IL17A levels after treatment were associated with long-term progression-free survival [100].

In addition, interactions between cytokines (e.g., IL-10 inhibiting the production of IL-1, IL-6, and IL-12) are complex and affect the overall immune response [101]. Different checkpoint inhibitors, such as the CTLA-4 and PD-1 blockade, may induce distinct immunologic changes and cytokine profiles [102]. Ignoring this temporal variability and interplay limits a full understanding of the role of cytokines as biomarkers. Single measurements at a single time point may not capture the complexity of the immune response and the dynamics of the tumor microenvironment. The dynamic monitoring of biomarkers is critical, especially in cancers [88,89]. Rather than relying on single measurements at a single time point, future studies should focus on profiling cytokines and other immune parameters in a serial manner to capture their kinetics and interplay during treatment. Such analysis can provide much more precise information about the response to therapy, early detection of resistance, and prediction of adverse events.

Understanding the nuanced roles of cytokines in the TME is critical for developing targeted immunotherapies that can enhance anti-tumor responses while minimizing immune-related adverse effects.

### 3.9. Eosinophiles

Eosinophils, best known for their role in allergic reactions and parasitic infections, are now increasingly recognized as influential players in the tumor microenvironment (TME) (Figure 4). Their involvement in cancer is complex and context-dependent, showing both pro- and anti-tumor effects, which vary with tumor type, stage, and immune context. This duality highlights the diverse roles eosinophils can play in cancer progression and the need to better understand the mechanisms behind their function in tumors. Once activated, eosinophils release a range of cytotoxic granules—including eosinophil peroxidase (EPO), major basic protein (MBP), eosinophil cationic protein (ECP), and eosinophil-derived neurotoxin (EDN)—that can directly induce tumor cell death. In addition, eosinophils secrete cytokines like IL-5, IL-13, and TNF-α, which influence the activity of other immune cells in the TME, such as T cells, natural killer cells, and macrophages. For instance, IL-5 enhances eosinophil recruitment, while IL-13 can affect macrophage polarization, and TNF-α may boosts T cell cytotoxicity. The impact of eosinophils on tumor outcome is highly variable. In cancers such as Hodgkin lymphoma and colorectal cancer, eosinophil infiltration has been linked to better prognosis and improved immune responses. In contrast, in lung cancer and melanoma, eosinophils have been associated with poor outcomes, potentially due to their role in immunosuppression or angiogenesis. Additionally, peripheral eosinophilia has been proposed as a positive predictive biomarker for response to immunotherapy, although it may also increase the risk of immune-related adverse events. The tumor microenvironment strongly influences eosinophil behavior. In pro-inflammatory settings, eosinophils may promote tumor destruction; in more immunosuppressive conditions, they may instead support tumor growth. As such, better insight into the signals guiding eosinophil recruitment and function is essential for leveraging their potential in cancer immunotherapy [103].

### 3.10. Tregs

Regulatory T cells (Tregs) are a subset of CD4^+^ T cells marked by Foxp3 expression that play a vital role in maintaining immune tolerance and preventing autoimmunity. However, in cancer, Tregs can suppress anti-tumor immune responses and support tumor progression by promoting an immunosuppressive tumor microenvironment (TME) [104,105].

Tregs inhibit the function of effector T cells, B cells, and NK cells through several mechanisms, notably the secretion of IL-10, TGF-β, and expression of CTLA-4. IL-10 dampens inflammatory cytokine production, while TGF-β suppresses T-cell proliferation and fosters further Treg differentiation. CTLA-4, by competing with CD28 for CD80/86 binding on antigen-presenting cells (APCs), limits co-stimulatory signals necessary for T-cell activation. Together, these actions foster immune tolerance and enable tumor cells to evade immune destruction. A high Treg presence within tumors has been correlated with poorer prognosis [105]. The balance between Tregs and effector T cells in the TME is a key factor influencing responses to immunotherapies [104]. A high Treg-to-effector T cell ratio typically indicates resistance to treatment, while a lower ratio is linked to improved outcomes. Accordingly, cancer therapies are exploring strategies to selectively deplete or inhibit Tregs within the tumor site [105]. However, because Tregs are essential for immune homeostasis, systemic depletion carries the risk of autoimmunity [104,105]. This underscores the need for precision in modulating Treg activity to enhance anti-tumor immunity while avoiding adverse effects.

### 3.11. MDSCs

Myeloid-derived suppressor cells (MDSCs) are a diverse group of immature myeloid cells that expand under pathological conditions such as cancer and chronic inflammation. These cells contribute to tumor progression by suppressing T-cell responses and shaping an immunosuppressive tumor microenvironment (TME). Normally rare in healthy individuals, MDSCs accumulate in malignancy due to cytokines and growth factors like IL-6, GM-CSF, and IL-10 that drive their differentiation and recruitment. MDSCs mediate immune suppression via multiple mechanisms: arginase (ARG1) depletes L-arginine, impairing T-cell function; inducible nitric oxide synthase (iNOS/NOS2) produces nitric oxide, which disrupts TCR signaling; and reactive oxygen species (ROS) contribute to oxidative stress and T-cell dysfunction. MDSCs also promote the expansion of regulatory T cells (Tregs) and M2-polarized macrophages, amplifying immunosuppression [106].

Their development and function are tightly regulated by signaling pathways such as JAK-STAT, NF-κB, and MAPK [94,95,96]. JAK-STAT signaling, triggered by IL-6 or GM-CSF, promotes MDSC differentiation and survival. NF-κB, activated by inflammatory stimuli like TNF-α and ROS, enhances the expression of immunosuppressive mediators. The MAPK pathway further supports MDSC development under stress or growth factor signaling. Conversely, AMPK signaling can dampen these pathways and limit MDSC expansion [107].

Studies indicate heterogeneity of MDSC populations and considerable interlaboratory variability in their phenotyping and quantification by flow cytometry, especially for granulocytic subsets [108]. The lack of a uniform, universally accepted classification of human MDSCs (both in terms of subset types and identification markers) and differences in gating strategies are the main sources of this variability [109]. This makes comparing study results and establishing reliable cutoff values for clinical assessment extremely difficult. Further efforts to harmonize protocols and standardize methodologies are essential for MDSCs to become routinely useful biomarkers in oncology [110].

Solving the heterogeneity problem in MDSC quantification is crucial. The complexity of MDSC subsets and the lack of consensus on the best combinations of surface markers for their identification by flow cytometry pose significant challenges. Standardization of gating strategies and protocols is essential to obtain consistent and comparable data, enabling their reliable clinical application.

Given their critical role in tumor immune evasion, MDSCs are increasingly recognized as promising targets in cancer immunotherapy.

### 3.12. Monocytes

Monocytes, a type of circulating white blood cell, are key players in the immune system, constantly surveilling the body for infections or injury. In solid tumors, these cells are actively recruited into the tumor microenvironment (TME) through chemokines, where they differentiate into macrophages [80]. This recruitment is a critical event that shapes the immune landscape of the TME and influences tumor progression [111,112] (Figure 4).

Once in the TME, macrophages adapt to environmental cues, leading to their polarization into distinct subtypes—primarily M1 and M2. M1 macrophages, activated by signals like IFN-γ and LPS, produce pro-inflammatory cytokines (e.g., TNF-α and IL-12) and reactive species that help eliminate tumor cells. In contrast, M2 macrophages, induced by IL-4, IL-10, and IL-13, support tumor growth, suppress immune responses, and facilitate tissue remodeling. These tumor-associated macrophages (TAMs) are typically skewed toward the M2 phenotype in tumors and are linked to metastasis, angiogenesis, and poor prognosis [112].

Cytokines within the TME strongly influence macrophage function. IL-6 promotes tumor growth and M2 polarization by enhancing immunosuppressive traits [112]. IL-10, a key anti-inflammatory cytokine, further pushes macrophages toward an M2 profile while inhibiting M1-mediated anti-tumor activity. Interestingly, TNF-α, though generally pro-inflammatory, can also drive M2 polarization depending on the context [112]. These cytokines, through their nuanced roles, shape the immunological behavior of macrophages and contribute to either tumor suppression or promotion.

Targeting macrophage polarization, especially reprogramming TAMs from an M2- to an M1-like state, is a promising therapeutic strategy aimed at enhancing anti-tumor immunity and disrupting tumor progression.

### 3.13. Mechanistic Insights into Blood Rheology Alterations and Their Impact on Tumor Biology and Immunotherapy Response

The relationship between changes in blood rheology and tumor biology, particularly in the context of responses to immunotherapy, is complex and multifaceted. Tumors often exhibit intrinsic resistance to immunotherapy, stemming from factors such as reduced lymphocyte activity against the tumor and mutations affecting immunological recognition [113]. For instance, mutations in genes associated with antigen presentation can limit the immune system’s ability to recognize tumor cells. Additionally, genomic instability, a hallmark of many cancers, can lead to the release of DNA into the cytoplasm, activating inflammatory pathways like the cGAS/STING pathway, potentially enhancing the anti-tumor immune response [114]. However, tumors often develop mechanisms to suppress these pathways to evade immune elimination, contributing to resistance to immunotherapy.

### 3.14. Inflammatory Signaling

The presence of DNA in the cytoplasm, resulting from genomic instability, activates DNA sensors like cGAS, leading to a type I interferon response that enhances anti-tumor immunity [114]. This inflammatory response can increase the immunogenicity of tumor cells, making them more susceptible to immunotherapies such as checkpoint inhibitors. However, tumors may develop mechanisms to suppress this signaling, for example, by inhibiting the cGAS/STING pathway, thereby limiting treatment efficacy. Furthermore, the tumor microenvironment can mimic fetal tissue, creating immunosuppressive conditions that hinder effective immune responses. Interactions between fetal-like macrophages and T cells play a crucial role in this process [115].

### 3.15. Tumor-Associated Inflammation and Blood Rheology

Cancer-related inflammation can also alter blood rheology. Studies have shown that patients with cancer exhibit increased plasma viscosity and changes in red blood cell aggregation, which are associated with inflammatory states [116]. These changes can affect blood flow within tumors, further complicating the delivery of immune cells and therapeutic agents.

### 3.16. Impact of Blood Rheology on the Tumor Microenvironment

In solid tumors, blood vessels are often abnormal, with dilated and tortuous structures, altering blood flow properties. Research indicates that blood viscosity within tumors is reduced compared to bulk viscosity, affecting tumor perfusion [117]. Impaired perfusion leads to hypoxia, a characteristic feature of the tumor microenvironment (TME), which is associated with immunosuppression. Hypoxia increases the expression of checkpoint molecules like PD-L1 on tumor and immune cells, promoting immune evasion [118]. Additionally, hypoxia attracts immunosuppressive cells, such as myeloid-derived suppressor cells (MDSCs) and regulatory T cells (Tregs), further inhibiting effective immune responses.

### 3.17. Strategies to Enhance Immunotherapy Responses

Strategies aimed at improving tumor perfusion, such as vascular normalization using anti-angiogenic drugs, can reduce hypoxia and enhance the efficacy of immunotherapy. Studies have demonstrated that administering low doses of anti-angiogenic drugs improves the delivery of immune and therapeutic cells to tumors, increasing the effectiveness of immunotherapy [119]. Sequential administration of these drugs can optimize the tumor-immune microenvironment, while high doses may lead to excessive vessel constriction and worsened perfusion. Similarly, anticoagulant therapy can promote the tumor’s immunological microenvironment and potentiate immunotherapy by alleviating hypoxia [120].

Understanding the immunosuppressive features of the tumor-immune microenvironment, such as VEGF-A expression, is crucial for developing strategies to improve responses to immunotherapy. Blocking specific pathways, for example, using anti-angiogenic drugs, can enhance treatment outcomes in cancers like hepatocellular carcinoma [115]. Changes in blood rheology, such as increased plasma viscosity, can limit the delivery of immune cells to tumors, reducing the effectiveness of therapies like checkpoint inhibitors or CAR-T cell therapy [116]. The normalization of blood flow strategies thus holds potential to increase the efficacy of immunotherapy, but it requires further research to optimize their clinical application.

### 3.18. Prognostic and Predictive Values of Four Blood- and Tumor-Based Biomarkers

In the presented review, the prognostic and predictive properties of biomarkers were discussed, with particular emphasis on their application in the context of immunotherapy. It is important to note that prognostic biomarkers provide information about the overall course of cancer regardless of the treatment applied, whereas predictive biomarkers offer insights into the likely response to a specific therapy. In the context of immunotherapy, combining prognostic markers such as NLR, PLR, and LMR with blood rheology parameters may enhance the ability to predict treatment outcomes. For instance, in a study of gastric cancer patients treated with immune checkpoint inhibitors, elevated NLR and PLR were associated with poorer survival outcomes, suggesting their potential as predictive biomarkers. Additionally, in the realm of blood rheology, changes such as increased plasma viscosity or alterations in erythrocyte aggregation can influence blood flow within the tumor, further complicating the delivery of immune cells and drugs. Understanding these interactions is crucial for optimizing cancer treatment strategies, considering both prognostic biomarkers and blood rheology parameters. These factors complement each other and should be applied individually depending on the clinical objective [18].

The prognostic and predictive value of blood markers such as the neutrophil-to-NLR, -PLR, and -LMR, has been extensively studied across various cancer types. Elevated NLR and PLR were generally associated with poorer OS and PFS, while higher LMR correlates with improved outcomes. However, their predictive value—specifically their ability to forecast response to specific treatments—is less established and appears to be context-dependent. For instance, in medullary thyroid carcinoma, LMR and MPV have shown potential in predicting postoperative calcitonin progression, but not recurrence [121].

It has been observed that elevated plasma and whole blood viscosity levels are linked to poorer survival and increased risk of metastasis in various cancers, including gynecologic and liver cancers [122]. Altered red blood cell aggregation and deformability can reflect disease severity and may serve as prognostic indicators, particularly in hematologic malignancies such as multiple myeloma. The mechanical properties of tumor tissues and cells—such as stiffness and cytoplasmic viscosity—are associated with treatment response and metastatic potential, providing opportunities for the development of predictive biomarkers.

These inflammatory markers have shown significant associations with patient outcomes, particularly in the context of ICIs and surgical interventions. These markers and their prognostic and predictive values are presented in Table 4 and Table 5.

## 4. Discussion and Conclusions

Based on the results discussed in the paper, there is a need for further analysis and searching for correlations between the value of whole blood viscosity and determinants that depend on the values of biochemical parameters and biomarkers. For example, the presence of neoplastic disease may alter factors governing whole blood viscosity, which may affect both blood circulation through normal and tumor tissues and even induce the onset of metastases. During cancer, changes in hematocrit values, changes in whole blood viscosity, plasma viscosity, and changes in erythrocyte aggregability and deformability were observed [41,50]. Han et al. emphasized in their studies that whole blood viscosity increases during cancer development and that it is associated with an advanced stage of systemic metastases [52]. In turn, R.K. Jain in 1988 already pointed out that there is a lack of quantitative data on the factors determining blood flow in tumors and that microcirculation in the tumor is heterogeneous both temporally and spatially. The author pointed out that answers to questions about intertumoral viscosity, how it changes depending on the size, type, and location of the tumor and whether the observed differences can be explained on the basis of the number, size, and stiffness of cells in the blood, plasma viscosity, and vessel geometry, and also what key parameters are needed to predict the hemodynamic response of human tumors should be found [44]. The results suggest the need for further large-scale studies, which may eventually enable the inclusion of cost-effective basic blood parameters in routine testing to evaluate immunotherapy treatment outcomes, especially regarding the lack of scientific research for the group of patients with solid tumors.

One of the most thoroughly studied peripheral blood biomarkers, though still lacking validation in large clinical trials, is soluble PD-L1 (sPD-L1). In the meta-analysis, Cheng et al. showed that low levels of sPD-L1 were significantly associated with improved OS and PFS [131]. However, further studies are needed to clearly determine whether sPD-L1 can serve as an independent prognostic factor in immunotherapy. Like soluble PD-L1 (sPD-L1), soluble CTLA-4 (sCTLA-4) has emerged as a potentially valuable biomarker for monitoring immunotherapeutic efficacy. Pistillo et al. reported significant associations between sCTLA-4 serum levels and ipilimumab response, OS, and irAEs [132]. An undeniable advantage of assessing parameters such as NLR, PLR, and LMR is their low cost, as they can be calculated from routinely performed complete blood counts with a differential. As previously mentioned, the most assessed peripheral blood biomarker remains ctDNA. A clear drawback of using such biomarkers is the variability in testing methods, protocols, and reporting standards [133,134].

While tissue-based biomarkers, such as PD-L1 expression, TMB, and MSI, are currently the most-often-used biomarkers for ICIs treatment, they require obtaining a tissue sample from the tumor. Immunohistochemical (IHC) assays for assessment of PD-L1 expression in tumor and immune cells were developed along PD-(L)1 inhibitors. Standardized assays include 22C3 and 28-8 pharmDx on Dako platforms, as well as SP142 and SP263 on Ventana platforms. The results of PD-L1 testing may vary depending on the monoclonal antibody, IHC platform, and interpreting pathologist. Studies showed that 28-8, 22C3, and SP263 assays were highly concordant for tumor cells staining, but not for immune cells staining [135]. A multicenter study by Adam et al. analyzed the concordance of the aforementioned standardized assays as well as 27 laboratory-developed tests, with combinations of five anti-PD-L1 monoclonal antibodies (28-8, 22C3, E1L3N, SP142, and SP263) and three types of IHC platforms (Dako Autostainer Link 48, Leica Bond, and Ventana BenchMark Ultra). A total of 14 of the laboratory-developed tests (51.8%) demonstrated similar concordance to the standardized assays [136].

Blood-based biomarkers have the advantage of being obtained with a simple blood draw instead of a biopsy. However, the diagnostic laboratory needs access to necessary equipment and qualified personnel. The cost and availability depends on the specific biomarker. NLR, LMR, and PLR can be calculated at any laboratory with a hematology analyzer as elements of a routine complete blood count. Variability can be minimized by quality control and analyzing fresh blood samples [137,138]. If necessary, samples can be sent to a reference laboratory, but optimal storage conditions should be provided [138,139]. Some rheological parameters of blood (such as hematocrit) can be assessed as part of the complete blood count, but analyzing erythrocyte deformability and whole blood viscosity requires additional equipment [140].

Combined with the often high cost of these analyses, this highlights the need to explore alternative, more affordable peripheral blood markers that could serve similar functions. It is worth emphasizing that even FDA-approved biomarkers show variable responses to ICIs therapy. A combination of several biomarkers may be required to enhance the accuracy of predicting therapeutic outcomes and optimize patient stratification [141]. The accumulation of myeloid-derived suppressor cells (MDSCs) is most prominent in patients with stage IV disease; however, elevated levels can already be observed in stage I [142]. Increased MDSC levels in gastrointestinal cancers have been identified as an independent prognostic factor for reduced overall survival, commonly associated with elevated concentrations of interleukin-13 (IL-13), arginase-1, and regulatory T cells (Tregs) [143]. Furthermore, a rising proportion and absolute number of MDSCs correlate with an increased metastatic tumor burden in patients with breast cancer [144] and colorectal cancer [145]. Elevated serum LDH levels have been consistently associated with poor prognosis in a variety of malignancies [146]. Elevated serum LDH levels are correlated with poor survival outcomes in solid tumors, particularly in melanoma, prostate, and renal cell carcinomas, and can serve as a valuable and cost-effective prognostic biomarker in metastatic carcinomas [147]. Cytokines exert both direct and indirect effects on tumor cell growth and metastatic behavior, as well as on stromal cells, including fibroblasts, infiltrating immune cells, and endothelial cells within the microvasculature [148]. Patients with metastatic solid tumors exhibited significantly higher mean serum IL-2R levels compared to those without metastases, with values similar to those observed in lymphoma patients [149].

Assay standardization and preanalytical variability are crucial for clinical use of biomarkers to ensure accurate and comparable results. Special attention is paid to ensure that the assays are robust to common confounding factors such as hemoglobin and bilirubin, which reduces preanalytical variability. However, detailed procedures such as sample storage are not fully controlled or may be a potential point of variability. Therefore, further studies are needed to establish universal standards for these biomarkers in clinical practice [150,151]. Key elements of standardization should include basic steps in the validation of assay methods, such as standard curve development, assay sensitivity, detection limits, quantification limits, assay repeatability, assay specificity, dilution linearity, and post-dose recovery. These elements ensure that assays are precise, reproducible, and suitable for clinical use, which is essential for the comparability of results between laboratories.

Preanalytical variability is one of the parameters that influence the obtained results and biomarker parameter values. Preanalytical variability refers to variations that may occur before sample analysis, such as sample collection, storage, and processing. For samples, steps such as interference testing, precise sample type, and sample handling, i.e., when biomarker measurement should be performed, whether before or after surgical interventions, are important. Although details such as storage conditions and processing time are not widely discussed, it is important to standardize sample collection procedures. The influence of tumor type and surgical intervention type is also important. Studies indicate the need for harmonization of biomarker measurement methods across laboratories. For example, other studies often use ELISA to measure these biomarkers, but without standardized protocols, results may be incomparable between laboratories. The need for common reference standards and sample handling protocols is crucial for the clinical application of these biomarkers. The accompanying publication fills a gap in the review by providing detailed information on assay standardization for sPD-L1 and sCTLA-4, which is crucial for their clinical application. It also addresses preanalytical variability, indicating that the assays are robust to common interferences, although detailed preanalytical procedures require further discussion. Further studies are needed to establish universal standards, which is necessary for the broad application of these biomarkers in clinical practice [152].

The diagnostic and prognostic values of MLR, NLR, PLR, CEA, and CA19-9 were assessed using a receiver operating characteristic (ROC) curve analysis, and the chi-square test in a retrospective analysis was conducted involving 783 patients diagnosed with CRC and 1232 age-matched healthy controls. Levels of MLR, NLR, and PLR were significantly elevated in CRC patients compared to controls. The areas under the ROC curve (AUC) for MLR, CEA, PLR, NLR, and CA19-9 were 0.739, 0.726, 0.683, 0.610, and 0.603, respectively. The combination of CEA and MLR yielded the highest diagnostic performance (AUC = 0.815), outperforming other combinations, including CEA with CA19-9. These findings indicate that MLR is a superior diagnostic marker for colorectal cancer compared to NLR and PLR, whereas NLR may hold greater prognostic value for CRC patients [153]. 

Another study highlighting the superiority of NLR and PLR over more costly diagnostic methods focused on gastric cancer. Most patients with gastric cancer remain asymptomatic until the disease reaches advanced stages. Although gastroscopy remains the gold standard diagnostic tool recommended by clinical guidelines, its invasive nature and high cost limit its widespread use in screening programs for early detection. Commonly used tumor markers, such as carcinoembryonic antigen (CEA) and carbohydrate antigen 19-9 (CA19-9), exhibit limited sensitivity and specificity, thereby restricting their diagnostic utility in gastric cancer. This retrospective study included 2606 patients diagnosed with gastric cancer (GC) and 3219 healthy controls monitored over a three-year period. Peripheral blood samples were analyzed to assess levels of NLR, PLR, CEA, and CA19-9. Optimal cutoff values for NLR and PLR were established through a receiver operating characteristic (ROC) curve analysis.

The findings demonstrated that systemic inflammatory markers NLR and PLR were significantly associated with gastric cancer diagnosis, particularly among male patients. These results suggest that assessment of inflammatory markers in peripheral blood may serve as a valuable tool for identifying high-risk populations for gastric cancer [154].

Temporal dynamics refer to the evolution of biomarkers, such as immune cell populations, gene expression profiles, or ctDNA, during immunotherapy.

Circulating non-coding RNAs (ncRNAs)—including microRNAs (miRNAs), long non-coding RNAs (lncRNAs), and circular RNAs (circRNAs)—are emerging as promising non-invasive biomarkers for cancer diagnosis and monitoring. These molecules are stable in body fluids such as blood and urine, making them suitable candidates for liquid biopsy applications. Their ability to reflect tumor-specific alterations underscores their potential in early detection, therapeutic selection, and disease monitoring.

miR-21, for instance, has been extensively studied in NSCLC. Elevated levels of miR-21 have been associated with tumor progression and poor prognosis, highlighting its utility as a diagnostic and prognostic biomarker [155].

In gastric cancer, the circular RNA hsa_circ_0001789 has demonstrated diagnostic value. Reduced expression levels of hsa_circ_0001789 in gastric cancer tissues and plasma samples correlate with malignant characteristics, suggesting its role in disease monitoring [156].

Moreover, ncRNAs have shown potential in predicting therapeutic responses. Certain ncRNA expression profiles can indicate sensitivity to targeted therapies, such as epidermal growth factor receptor (EGFR) inhibitors in NSCLC, facilitating personalized treatment strategies [157].

Unlike static pretreatment measurements, longitudinal assessments capture the immune system’s response to treatment, providing insights into mechanisms of action and clinical outcomes. These dynamics are particularly relevant for PD-1 and CTLA-4 inhibitors, which modulate T-cell activity at different stages, PD-1 primarily in peripheral tissues later in the immune response, and CTLA-4 in lymph nodes early on [158].

PD-1 and CTLA-4 inhibitors target distinct immune checkpoints, leading to different temporal biomarker profiles. PD-1 blockade primarily enhances exhausted CD8+ T-cell proliferation early in treatment, as seen in lung cancer patients where PD-1+CD8+ T cells peaked within the first week [152]. Conversely, CTLA-4 blockade promotes CD4+ Th1-like effector cell expansion and Treg depletion, with effects more evident at later time points (e.g., 3 weeks post-treatment) [159]. Combination therapy synergizes these effects, sustaining robust T-cell responses, as evidenced by increased Ki-67+ CD8+ T cells in peripheral blood post-ipilimumab plus nivolumab [160]. 

Implementing longitudinal biomarker analysis faces challenges, including the invasiveness of serial biopsies and the need for standardized timing protocols. Optimal timing for assessment varies by biomarker and treatment; for instance, PD-L1 IHC requires staining within 6 months of tissue sectioning, and T-cell assays need prompt PBMC isolation to avoid suppression [161]. Advances in liquid biopsies and high-throughput technologies, like mass cytometry and single-cell sequencing, could facilitate non-invasive, frequent monitoring. Future research should focus on validating optimal time points (e.g., 1, 3, or 6 weeks) and developing combination biomarker strategies to enhance predictive accuracy.

The temporal dynamics of biomarkers during a PD-1 and CTLA-4 blockade provide critical insights into immune responses, enabling improved patient stratification, response monitoring, and resistance management. Longitudinal studies demonstrate distinct patterns between PD-1 and CTLA-4 therapies, with combination approaches amplifying these effects. By integrating serial biomarker assessments into clinical practice, immunotherapy can be optimized, paving the way for precision oncology.

The use of multiparametric predictive models in cancer treatment, particularly in the context of immunotherapy, is justified for several reasons based on the analysis of the presented text and general trends in oncology. Integrating immunological markers such as CTLA-4 and PD-L1, inflammatory markers (NLR, PLR, and LMR), and rheological parameters such as RDW-SD, MCV, PDW, APTT, P-LCR, and MPV enables more accurate prediction of treatment response and identification of patients most likely to benefit from immune checkpoint inhibitor therapy.

Traditional biomarkers, such as PD-L1 expression, do not always provide sufficient predictive accuracy due to variability in detection methods, interpretation criteria, and selection thresholds. Multiparametric models, such as the random forest (RF) model, which incorporates a variety of blood parameters, e.g., RDW-SD, MCV, PDW, CD3+CD8+, APTT, P-LCR, calcium, MPV, CD4+/CD8+ ratio, and AST, demonstrate superior predictive performance compared to conventional models.

The integration of these markers allows for a holistic assessment of the patient’s condition, taking into account the immune response as well as inflammatory and rheological changes associated with cancer. For instance, the CD4+/CD8+ ratio is a key indicator of immune function; an elevated ratio suggests enhanced helper T cell activity relative to suppressor T cells, which correlates with a better prognosis in immunotherapy. Inflammatory markers such as RDW-SD are associated with inflammatory responses and oxidative stress, both of which influence tumor progression, while rheological parameters like APTT and MPV reflect changes in coagulation and platelet activation systems that may promote tumor development and affect treatment outcomes.

Using routine blood tests as a data source for predictive models is minimally invasive, cost-effective, and enables sequential patient monitoring. The RF model mentioned facilitates the stratification of patients into high- and low-risk groups, thereby assisting clinicians in identifying individuals most likely to benefit from immunotherapy, supporting therapeutic decision-making, and reducing the risk of ineffective treatment [162].

In this study, we emphasize the need to broaden the scope of current investigations to include additional systemic inflammatory markers, such as the NLR, PLR, and LMR, as well as hemorheological parameters. Although the dynamics of these indices during immunotherapy remain insufficiently characterized, they may hold significant clinical relevance and offer a cost-effective means to enhance the monitoring and personalization of immunotherapeutic strategies.

## Figures and Tables

**Figure 1 cancers-17-02001-f001:**
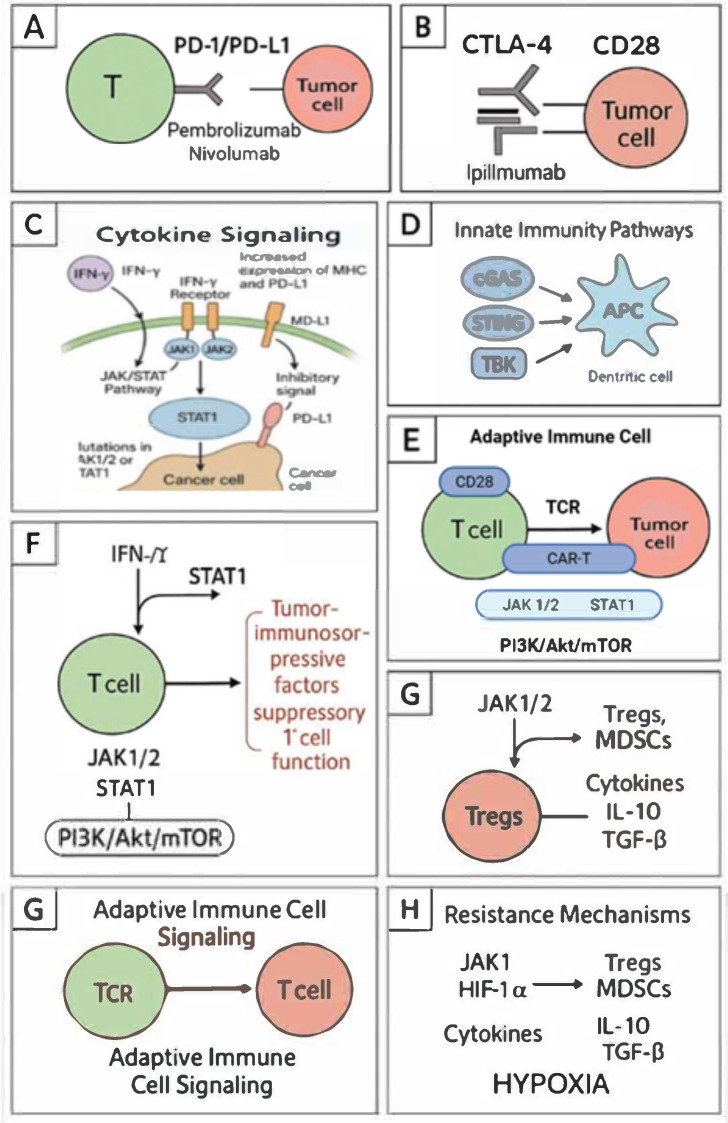
Key signaling pathways in cancer development and cancer immunotherapy. In (**A**), PD-1 on T cells binds PD-L1 on tumor cells; PD-1/PD-L1 inhibitors (nivolumab and pembrolizumab) block this interaction. In (**B**), CTLA-4 competes with CD28 for CD80 on antigen-presenting cells, inhibiting T cell activation; ipilimumab blocks CTLA-4. In (**C**), IFN-γ activates the JAK/STAT pathway, increasing MHC and PD-L1 expression on tumor cells; JAK1/2 or STAT1 mutations can cause immunotherapy resistance. In (**D**), cGAS detects cytosolic DNA, produces cGAMP, and activates STING, which induces type I interferons via TBK1 and IRF3, promoting dendritic cell maturation and T cell activation. In (**E**), TCR with MHC and CD28 activates the PI3K/Akt/mTOR pathway, enhancing T cell function; CAR-T therapy amplifies this effect. In (**F**), JAK1/2 mutations impair IFN-γ signaling, reducing MHC-I and PD-L1 expression, leading to PD-1 blockade resistance. PTEN loss activates PI3K, creating an immunosuppressive tumor microenvironment. Tumor cells inhibit PI3K/Akt/mTOR in T cells, impairing their function. In (**G**), TCR engagement with antigen-MHC and co-stimulation (e.g., CD28) activates PI3K/Akt/mTOR, promoting T cell survival, proliferation, and effector functions. In (**H**), tumor hypoxia upregulates HIF-1α, driving immunosuppressive pathways and reducing immune cell infiltration. MDSCs and Tregs suppress immunity via ARG1, iNOS, and TGF-β signaling. https://BioRender.com/quskbqk.

**Figure 2 cancers-17-02001-f002:**
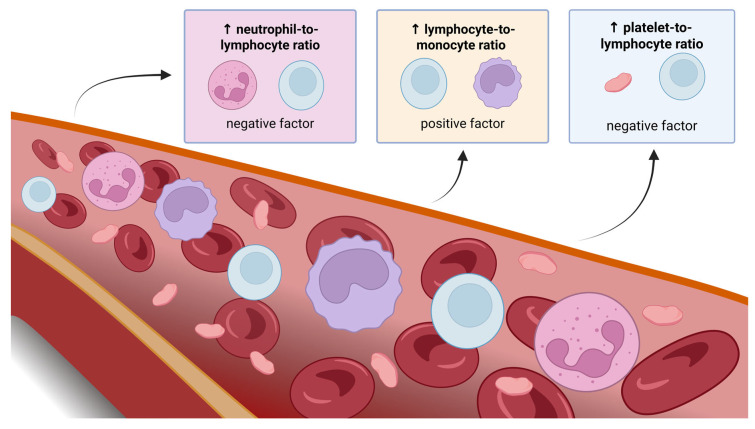
Peripheral blood cell ratios as predictive and prognostic indicators during immunotherapy. https://BioRender.com/buzdtfb.

**Figure 3 cancers-17-02001-f003:**
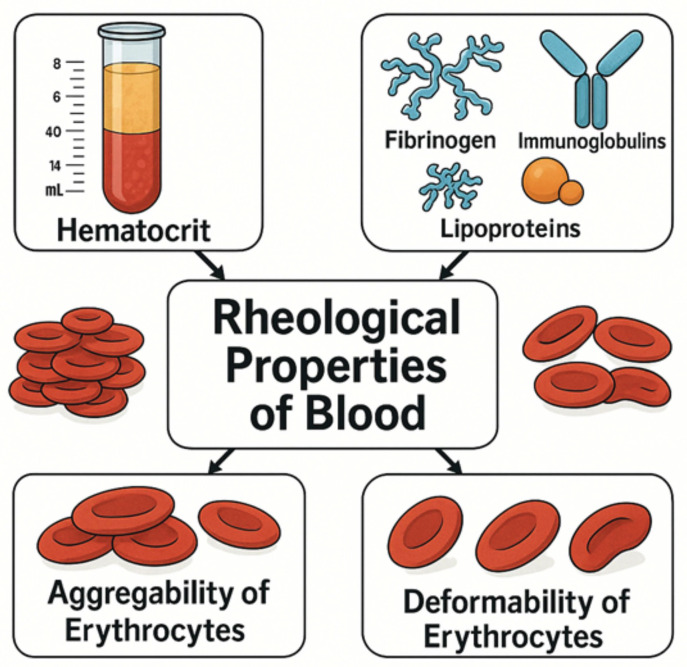
Rheological properties of blood. https://BioRender.com/3jagiky.

**Figure 4 cancers-17-02001-f004:**
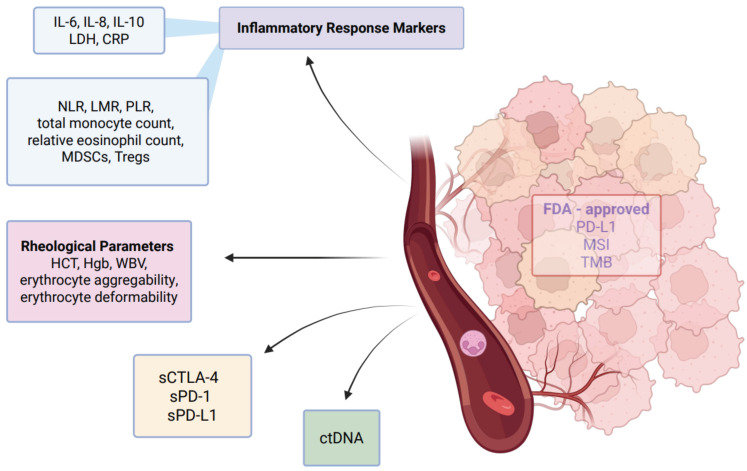
Blood- and tumor-based biomarkers for assessing responses to cancer immunotherapy. https://BioRender.com/zbddbm9.

**Table 1 cancers-17-02001-t001:** Rheological parameters and outcomes of immunotherapy in cancer.

Rheological Parameter	Cancer Type/Treatment	Association with Treatment Outcomes	Statistical Data	Reference
Whole blood viscosity (WBV)	HCC and treated with nivolumab	A higher WBV (>16.0 cP) was associated with worse OS and PFS.	OS: *p* = 0.069; PFS: *p* = 0.067; ORR: 25% (low WBV) vs. 0% (high WBV), *p* = 0.409	[52]
Hemoglobin (Hb)	Non-small-cell lung cancer (NSCLC) and immunotherapy	Hb ≥ 110 g/L associated with better OS and PFS	OS: 17.6 vs. 10.5 months, HR 0.56, *p* < 0.001; PFS: 10.0 vs. 4.0 months, HR 0.63, *p* = 0.001	[54]
Hemoglobin (Hb)	NSCLC and ICI	Higher baseline Hb associated with better OS and PFS	Detailed stats not disclosed; trend observed in 224 patients	[55]
Red cell-based rheologic score (Hb ≥ 12 g/dL, MCV > 87 fL, and RDW ≤ 16%)	Metastatic renal cell carcinoma (mRCC), treated with TKIs and/or ICIs	Higher score correlated with better OS and PFS	OS: 42.0 vs. 17.3 months, HR 0.60, *p* < 0.001; PFS: 17.4 vs. 8.2 months, HR 0.66, *p* < 0.001	[58]

**Table 2 cancers-17-02001-t002:** Preclinical studies assessing blood-based biomarkers in mouse models of cancer immunotherapy.

Animal Model	Tumor Subsite/Biomarkers	Immunotherapy	Biomarkers	Key Findings	Reference
mouse (KP lung adenocarcinoma and MC38)	lung adenocarcinoma and colon carcinoma	not specified	neutrophil gene signatures and, likely, blood neutrophil counts	Therapy-elicited neutrophils with interferon gene signatures are essential for successful immunotherapy.	[65]
mouse (various strains: BL/6 and BALB/c)	TC-1, CT26, B16F10, MC38, and 4T1	antimicrobial peptides and vaccination	blood neutrophil percentages	Neutrophil abundance and phenotype vary with host genetics and tumor type, affecting tumor growth control.	[66]
mouse	not specified	TNF, CD40 agonist, and tumor-binding antibody	neutrophil activation and infiltration	Neutrophils induced tumor eradication through oxidative damage.	[67]
humanized mouse (MISTRGGR)	not specified	not specified	human neutrophil counts	improved reconstitution of human neutrophils, enabling potential NLR calculation	[68]
PBMC humanized (NSG, etc.)	-	TGN1412 analog, a CD28 superagonist	cytokine levels (e.g., IL-6)	robust cytokine release in response to CD28 superagonist; useful for CRS assessment	[69]
HIS-BRGS mice	breast, colorectal, pancreatic, lung, adrenocortical, melanoma, and hematological malignancies	block CTLA-4 and/or PD-1/PD-L1	human chimerism and T cell subsets	correlated blood immune profiles with tumor infiltration; variable chimerism noted	[70]

**Table 3 cancers-17-02001-t003:** Selected original studies and their findings on blood-based biomarkers in immunotherapy for solid tumors.

Clinical Study Type—Phase	Tumor Subsite	Biomarkers	Immunotherapy	Key Findings	Reference
retrospective study	NSCLC	CD4/CD8, LYM%, PD-1+ T-cells, NLR, and MLR	anti-PD-(L)1	Elevated expression of CD4/CD8 and LYM% are positively associated with effective immunotherapy, while PD-1+ on T cells, NLR, and MLR have a negative impact.	[71]
phase 3 trial	NSCLC	bTMB	atezolizumab	Additional exploration of bTMB to identify optimal cutoffs, confounding factors, assay improvements, or cooperative biomarkers is warranted.	[72]
phase 2 and phase 3 trials	NSCLC	bTMB	atezolizumab	bTMB identifies patients who derive clinically significant improvements in PFS from atezolizumab.	[73]
multiple cohorts	NSCLC	CD4/CD8, LYM%, PD-1+ T-cells, NLR, and MLR	anti-PD-(L)1	ctDNA-adjusted bTMB might predict OS benefit in NSCLC patients receiving ICIs.	[74]
multiple cohorts	NSCLC	bTMB	anti-PD-(L)1	Ma-bTMB could reduce the confounding effect of MSAF and ITH on bTMB calculation and effectively identify beneficiaries of ICIs.	[75]
multiple cohorts	NSCLC	bTMB	anti-PD-(L)1	LAF-bTMB is a feasible predictor of OS, PFS, and ORR.	[76]
prospective cohort study	NSCLC	ctDNA-adjusted bTMB	anti-PD-(L)1	Presence of CTCs is a predictive factor for a worse durable response rate to ICIs.	[77]
prospective cohort study	melanoma	MSAF-adjusted bTMB	pembrolizumab	Early on-treatment increase in circulating exosomal PD-L1 stratifies clinical responders from nonresponders.	[78]
prospective cohort study	melanoma	Allele frequency-adjusted bTMB	ipilimumab	Increased exosomal PD-1 and CD28 on T-cells were correlated with longer PFS and OS.	[79]

**Table 4 cancers-17-02001-t004:** Prognostic and predictive value of NLR, PLR, and LMR in various cancers.

Cancer Type	NLR Prognostic Value	PLR Prognostic Value	LMR Prognostic Value	Reference
Gastric Cancer (ICI)	Elevated NLR associated with poorer OS (HR = 2.01) and PFS (HR = 1.59)	Elevated PLR associated with poorer OS (HR = 1.57) and PFS (HR = 1.52)	Elevated LMR associated with improved OS (HR = 0.62) and PFS (HR = 0.69)	[18]
Melanoma (ICI)	Elevated NLR associated with poorer OS and PFS	Elevated PLR associated with poorer OS and PFS	Elevated LMR associated with improved OS and PFS	[19]
Head and Neck SCC	Elevated NLR identified as an independent negative prognostic factor for OS	PLR not specified as a significant prognostic factor	LMR not specified as a significant prognostic factor	[123]
Breast Cancer	Elevated NLR correlated with poorer DSS and DFS	PLR identified as an independent prognostic marker with superior predictive value for DSS and DFS compared to NLR and LMR	Lower LMR associated with poorer DSS and DFS	[124]
Pancreatic Cancer	Elevated NLR associated with worse OS	Elevated PLR correlated with greater tumor viability post-neoadjuvant chemotherapy	LMR not significant as a prognostic marker	[125]
Osteosarcoma	Elevated NLR significantly correlated with advanced disease stage and poorer prognosis	Elevated PLR significantly correlated with advanced disease stage and poorer prognosis	Lower LMR significantly correlated with advanced disease stage and poorer prognosis	[126]
Laryngeal Carcinoma	Elevated NLR associated with increased mortality	Elevated PLR associated with increased mortality	Lower LMR associated with better survival outcomes	[121]
Hilar Cholangiocarcinoma	NLR negatively correlated with CD3^+^ and CD8^+^ TILs; associated with poorer OS	PLR showed no correlation with TILs	Elevated LMR positively correlated with CD3^+^ TILs; identified as an independent prognostic factor for OS	[127] Początek formularza
				Dół formularza

**Table 5 cancers-17-02001-t005:** Prognostic and predictive value of rheological parameters in cancer.

**Cancer Type**	**Rheological Parameter(s)**	**Prognostic/Predictive Value**	**Reference**
Gynecologic Cancers	Plasma viscosity and RBC aggregation	Elevated plasma viscosity is an independent prognostic marker for overall survival in breast and ovarian cancers; higher plasma viscosity correlates with increased risk of thrombosis and poorer survival outcomes.	[122]
Hepatocellular Carcinoma	Whole blood viscosity	Increased whole blood viscosity is associated with extrahepatic metastases and reduced survival, indicating its potential as a prognostic marker.	[52]
Colorectal Liver Metastases	Tissue stiffness (shear wave speed) and viscoelastic parameters (α and µ)	Higher tissue stiffness correlates with better histopathological response to chemotherapy; viscoelastic parameters can predict treatment response with high diagnostic accuracy (AUC > 0.8).	[128]
Multiple Myeloma	RBC aggregation index and deformability	Patients exhibit higher RBC aggregation and reduced deformability compared to healthy controls, which may contribute to disease progression and could serve as prognostic indicators.	[46]
Various Cancers (Pre/Post-Chemotherapy)	Hematocrit, ESR, plasma viscosity, and whole blood viscosity	Chemotherapy induces significant changes in rheological parameters; post-chemotherapy reductions in whole blood viscosity and hematocrit may reflect treatment response and impact prognosis.	[129]
Breast Cancer (Cellular Level)	Cytoplasmic viscosity	Lower cytoplasmic viscosity in highly metastatic breast cancer cells suggests its potential as a biomarker for metastatic potential and aggressiveness.	[130]
Colon Cancer	Tissue rheology (compressional stiffening and shear weakening)	Cancerous colon tissues exhibit distinct rheological properties compared to healthy tissues; these mechanical characteristics may serve as complementary diagnostic markers alongside histopathology.	[131]

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
