# Peer review of "Blood-Based Biomarkers as Predictive and Prognostic Factors in Immunotherapy-Treated Patients with Solid Tumors—Currents and Perspectives"

_cancers, 2025, doi:10.3390/cancers17122001_

Round 1
Reviewer 1 Report
Comments and Suggestions for Authors
Some of the suggestions that could improve the quality of this work include:
- Despite a plethora of blood-based biomarkers being listed in the review, they are neither harmonized nor quantified in terms of their predictive or prognostic strength, restricting our ability to determine their clinical utility.
- The authors present various rheological parameters like whole blood viscosity as new biomarkers in monitoring the efficacy of immunotherapy, but do not provide data or statistics linking these parameters to treatment outcomes as stated in the title of this review.
- Despite the further common statement of the mechanisms for linking Blood Rheology Alteration to Tumor Biology and Response to Immunotherapy, this is still joint without mechanistic details; Here, a more precise hsfrom needs to be addressed.
- However, in one instance, predictive (as opposed to prognostic) roles of individual biomarkers such as NLR, LMR and PLR are mentioned; whereas the predictive and prognostic terms are used interchangeably throughout the text and there is no clear delineation between these two terms.
- While the review periodically discusses the usefulness of soluble checkpoint molecules such as sPD-L1 and sCTLA-4, there is a lack of discussion regarding assay standardization or pre-analytical variability, both of which are essential for clinical implementation.
- The review refers to levels of cytokines, such as IL-6, IL-8, IL-10, but does not consider either intra-patient temporal variation or the relationships between these cytokines and other immune parameters over the course of therapy.
- While the current text quotes a meta-analysis linking low levels of sPD-L1 to superior outcomes, it falls short of providing cohort size, inclusion criteria, or statistical significance to qualify its findings.
- The addition of myeloid-derived suppressor cells (MDSCs) as a biomarker is beneficial but the review could have incorporated harmonized strategies for their detection and quantification which is a major hurdle to clinical application.
- The review does not consider that ctDNA, while the most studied blood biomarker/marker, may not always adequately reflect tumor heterogeneity or clonal evolution, factors that similarly affect detection and interpretation over time.
- LDH is inexpensive and widely available as a prognostic marker as mentioned in this review but it has not been compared for its sensitivity or specificity against other systemic inflammation markers.
- While there are calls for multi-biomarker combinations to increase predictive power, this review does not provide a framework or risk-scoring system for integrating the different biomarkers into a single risk stratification system.
- The review has references, highlighting the cost effectiveness of the NLR, PLR and LMR, but no comparative performance metrics or ROC analyses are provided to show that these are better than a more sophisticated (but costly) test.
- However, without further clinical trial data supporting the claim that rheological parameters can become a new standard in the monitoring of immunotherapy, this statement seems speculative.
- Therefore, due to lack of statistical meta-analyses or pooled hazard ratios to substantiate the prognostic significance of the biomarkers addressed in this systematic review, the strength of the clinical recommendations mentioned will be limited.
- That aside, the review points out that blood-based markers are easier for both patients and clinicians to access in comparison, but fails to mention differences in factors such as availability of assays, calibration or interpretation in clinical laboratories.
- The ability of immunotherapy-induced immune-related adverse events (irAEs) to confound interpretation of specific systemic biomarkers is not addressed, either.
- Although it references historical studies on the micro-circulation and viscosity of tumors, no recent imaging or computational modeling studies that quantify those observations appear to be available.
- These studies do not report the impact of baseline comorbidities (e.g. CV disease, diabetes) on blood-based biomarker levels, which could alter their population specificity in the cancer context.
- Although insights into the temporal dynamics of these biomarkers are critical to improving their clinical utility (e.g., during PD-1 vs. CTLA-4 blockade) they are not explicitly addressed.
- Although the review appropriately notes that most of the biomarkers discussed are unvalidated in large-scale studies, it does not set out specific appropriate trial designs or biomarker-driven endpoints that could direct future research
Author Response
Dear Reviewer,
We sincerely thank you for their valuable comments and suggestions, which have helped us improve the clarity and quality of our manuscript. All responses to the reviewer’s remarks have been addressed in the revised text and are highlighted in a different color for ease of reference.
- Despite a plethora of blood-based biomarkers being listed in the review, they are neither harmonized nor quantified in terms of their predictive or prognostic strength, restricting our ability to determine their clinical utility.
We are grateful for this thoughtful and constructive suggestion.
The prognostic and predictive value of blood markers such as the neutrophil-to-NLR, PLR and LMR has been extensively studied across various cancer types. Elevated NLR and PLR were generally associated with poorer overall survival (OS) and progression-free survival (PFS), while higher LMR correlates with improved outcomes. However, their predictive value—specifically their ability to forecast response to specific treatments—is less established and appears to be context-dependent. For instance, in medullary thyroid carcinoma, LMR and mean platelet volume (MPV) have shown potential in predicting postoperative calcitonin progression, but not recurrence [Li C, Zhang H, Li S, Zhang D, Li J, Dionigi G, Liang N, Sun H. Prognostic Impact of Inflammatory Markers PLR, LMR, PDW, MPV in Medullary Thyroid Carcinoma. Front Endocrinol (Lausanne). 2022 Mar 8;13:861869. doi: 10.3389/fendo.2022.861869. PMID: 35350101; PMCID: PMC8957807].
It has been observed that elevated plasma and whole blood viscosity levels are linked to poorer survival and increased risk of metastasis in various cancers, including gynecologic and liver cancers [von Tempelhoff GF, Nieman F, Heilmann L, Hommel G. Association between blood rheology, thrombosis and cancer survival in patients with gynecologic malignancy. Clin Hemorheol Microcirc. 2000;22(2):107-30. PMID: 10831062.]. Altered red blood cell aggregation and deformability can reflect disease severity and may serve as prognostic indicators, particularly in hematologic malignancies such as multiple myeloma. The mechanical properties of tumor tissues and cells—such as stiffness and cytoplasmic viscosity—are associated with treatment response and metastatic potential, providing opportunities for the development of predictive biomarkers.
These inflammatory markers have shown significant associations with patient outcomes, particularly in the context of immune checkpoint inhibitors (ICIs) and surgical interventions. These markers and their prognostic and predictive values were presented in Table presented in the revised version of our manuscript.
- The authors present various rheological parameters like whole blood viscosity as new biomarkers in monitoring the efficacy of immunotherapy, but do not provide data or statistics linking these parameters to treatment outcomes as stated in the title of this review.
We thank you for this valuable and insightful comment.
In a clinical study involving 33 patients with hepatocellular carcinoma (HCC) treated with nivolumab, a PD-1 inhibitor, high diastolic whole blood viscosity (WBV) (>16.0 cP) was associated with poorer overall survival (OS) and progression-free survival (PFS) compared to the group with lower WBV (≤16.0 cP). The differences were close to statistical significance (P=0.069 for OS, P=0.067 for PFS), which may be due to the small sample size. The objective response rate was 25.0% in the low WBV group (6/24, including 1 complete response and 5 partial responses) compared to 0% in the high WBV group (0/9), although the difference was not statistically significant (P=0.409). The study suggests that WBV may serve as a potential biomarker in HCC, particularly in the context of immunotherapy, but emphasizes the need for larger, prospective studies. [Han JW, Sung PS, Jang JW, Choi JY, Yoon SK. Whole blood viscosity is associated with extrahepatic metastases and survival in patients with hepatocellular carcinoma. PLoS One. 2021 Dec 2;16(12):e0260311. doi: 10.1371/journal.pone.0260311. PMID: 34855786; PMCID: PMC8638904].
The Table in our revised manuscript presents the available data on the impact of blood rheological parameters on immunotherapy outcomes across various cancer types. In some cases, although certain trends were observed, the results did not reach statistical significance, which may be attributed to the small sample sizes of the study populations.
- Despite the further common statement of the mechanisms for linking Blood Rheology Alteration to Tumor Biology and Response to Immunotherapy, this is still joint without mechanistic details; Here, a more precise hsfrom needs to be addressed.
We sincerely appreciate your careful reading and helpful feedback.
Mechanistic insights into blood rheology alterations and their impact on tumor biology and immunotherapy response. The relationship between changes in blood rheology and tumor biology, particularly in the context of responses to immunotherapy, is complex and multifaceted. Tumors often exhibit intrinsic resistance to immunotherapy, stemming from factors such as reduced lymphocyte activity against the tumor and mutations affecting immunological recognition [Rieth J, Subramanian S. Mechanisms of Intrinsic Tumor Resistance to Immunotherapy. Int J Mol Sci. 2018 May 2;19(5):1340. doi: 10.3390/ijms19051340. PMID: 29724044; PMCID: PMC5983580]. For instance, mutations in genes associated with antigen presentation can limit the immune system's ability to recognize tumor cells. Additionally, genomic instability, a hallmark of many cancers, can lead to the release of DNA into the cytoplasm, activating inflammatory pathways like the cGAS/STING pathway, potentially enhancing the anti-tumor immune response [Chen M, Linstra R, van Vugt MATM. Genomic instability, inflammatory signaling and response to cancer immunotherapy. Biochim Biophys Acta Rev Cancer. 2022 Jan;1877(1):188661. doi: 10.1016/j.bbcan.2021.188661. Epub 2021 Nov 17. PMID: 34800547]. However, tumors often develop mechanisms to suppress these pathways to evade immune elimination, contributing to resistance to immunotherapy.
Inflammatory Signaling
The presence of DNA in the cytoplasm, resulting from genomic instability, activates DNA sensors like cGAS, leading to a type I interferon response that enhances anti-tumor immunity [Chen et al., 2022]. This inflammatory response can increase the immunogenicity of tumor cells, making them more susceptible to immunotherapies such as checkpoint inhibitors. However, tumors may develop mechanisms to suppress this signaling, for example, by inhibiting the cGAS/STING pathway, thereby limiting treatment efficacy. Furthermore, the tumor microenvironment can mimic fetal tissue, creating immunosuppressive conditions that hinder effective immune responses. Interactions between fetal-like macrophages and T cells play a crucial role in this process [Jennifer Currenti, Archita Mishra, Michael Wallace, Jacob George, Ankur Sharma; Immunosuppressive mechanisms of oncofetal reprogramming in the tumor microenvironment: implications in immunotherapy response. Biochem Soc Trans 26 April 2023; 51 (2): 597–612. doi: https://doi.org/10.1042/BST20220157].
Tumor-Associated Inflammation and Blood Rheology
Cancer-related inflammation can also alter blood rheology. Studies have shown that patients with cancer exhibit increased plasma viscosity and changes in red blood cell aggregation, which are associated with inflammatory states [Tikhomirova I, Petrochenko E, Muravyov A, et al. Microcirculation and blood rheology abnormalities in chronic heart failure. Clinical Hemorheology and Microcirculation. 2016;65(4):383-391. doi:10.3233/CH-16206]. These changes can affect blood flow within tumors, further complicating the delivery of immune cells and therapeutic agents.The Open Neurology Journal+8Heart Lung Circulation+8journals.rcsi.science+8
Impact of Blood Rheology on the Tumor Microenvironment
In solid tumors, blood vessels are often abnormal, with dilated and tortuous structures, altering blood flow properties.Research indicates that blood viscosity within tumors is reduced compared to bulk viscosity, affecting tumor perfusion [Sevick EM, Jain RK. Geometric resistance to blood flow in solid tumors perfused ex vivo: effects of tumor size and perfusion pressure. Cancer Res. 1989 Jul 1;49(13):3506-12. PMID: 2731172]. Impaired perfusion leads to hypoxia, a characteristic feature of the tumor microenvironment (TME), which is associated with immunosuppression. Hypoxia increases the expression of checkpoint molecules like PD-L1 on tumor and immune cells, promoting immune evasion [Noman MZ, Parpal S, Van Moer K, et al. Inhibition of Vps34 reprograms cold into hot inflamed tumors and improves anti-PD-1/PD-L1 immunotherapy. Sci Adv. 2020 Apr 29;6(18):eaax7881. doi: 10.1126/sciadv.aax7881. Erratum in: Sci Adv. 2021 Apr 9;7(15):eabf5801. doi: 10.1126/sciadv.abf5801. PMID: 32494661; PMCID: PMC7190323]. Additionally, hypoxia attracts immunosuppressive cells, such as myeloid-derived suppressor cells (MDSCs) and regulatory T cells (Tregs), further inhibiting effective immune responses.digitalcommons.bucknell.edu
Strategies to Enhance Immunotherapy Responses
Strategies aimed at improving tumor perfusion, such as vascular normalization using anti-angiogenic drugs, can reduce hypoxia and enhance the efficacy of immunotherapy. Studies have demonstrated that administering low doses of anti-angiogenic drugs improves the delivery of immune and therapeutic cells to tumors, increasing the effectiveness of immunotherapy [Mpekris F, Voutouri C, Baish JW, et al. Combining microenvironment normalization strategies to improve cancer immunotherapy. Proc Natl Acad Sci U S A. 2020 Feb 18;117(7):3728-3737. doi: 10.1073/pnas.1919764117. Epub 2020 Feb 3. PMID: 32015113; PMCID: PMC7035612]. Sequential administration of these drugs can optimize the tumor-immune microenvironment, while high doses may lead to excessive vessel constriction and worsened perfusion. Similarly, anticoagulant therapy can promote the tumor's immunological microenvironment and potentiate immunotherapy by alleviating hypoxia [Kim R, An M, Lee H, et al. Early Tumor-Immune Microenvironmental Remodeling and Response to First-Line Fluoropyrimidine and Platinum Chemotherapy in Advanced Gastric Cancer. Cancer Discov. 2022 Apr 1;12(4):984-1001. doi: 10.1158/2159-8290.CD-21-0888. PMID: 34933901; PMCID: PMC9387589].
Understanding the immunosuppressive features of the tumor-immune microenvironment, such as VEGF-A expression, is crucial for developing strategies to improve responses to immunotherapy. Blocking specific pathways, for example, using anti-angiogenic drugs, can enhance treatment outcomes in cancers like hepatocellular carcinoma [Currenti et al., 2023].Changes in blood rheology, such as increased plasma viscosity, can limit the delivery of immune cells to tumors, reducing the effectiveness of therapies like checkpoint inhibitors or CAR-T cell therapy [Tikhomirova et al., 2016]. Normalization of blood flow strategies thus hold potential to increase the efficacy of immunotherapy but require further research to optimize their clinical application.
- However, in one instance, predictive (as opposed to prognostic) roles of individual biomarkers such as NLR, LMR and PLR are mentioned; whereas the predictive and prognostic terms are used interchangeably throughout the text and there is no clear delineation between these two terms.
We thank you for raising this important point, which allowed us to clarify the scope of our review. In the presented review, the prognostic and predictive properties of biomarkers were discussed, with particular emphasis on their application in the context of immunotherapy. It is important to note that prognostic biomarkers provide information about the overall course of cancer regardless of the treatment applied, whereas predictive biomarkers offer insights into the likely response to a specific therapy. In the context of immunotherapy, combining prognostic markers such as NLR, PLR, and LMR with blood rheology parameters may enhance the ability to predict treatment outcomes. For instance, in a study of gastric cancer patients treated with immune checkpoint inhibitors, elevated NLR and PLR were associated with poorer survival outcomes, suggesting their potential as predictive biomarkers. Additionally, in the realm of blood rheology, changes such as increased plasma viscosity or alterations in erythrocyte aggregation can influence blood flow within the tumor, further complicating the delivery of immune cells and drugs. Understanding these interactions is crucial for optimizing cancer treatment strategies, considering both prognostic biomarkers and blood rheology parameters. These factors complement each other and should be applied individually depending on the clinical objective [Tan S, Zheng Q, Zhang W, Zhou M, Xia C, Feng W. Prognostic value of inflammatory markers NLR, PLR, and LMR in gastric cancer patients treated with immune checkpoint inhibitors: a meta-analysis and systematic review. Front Immunol. 2024 Jul 10;15:1408700. doi: 10.3389/fimmu.2024.1408700. PMID: 39050856; PMCID: PMC11266030].
- While the review periodically discusses the usefulness of soluble checkpoint molecules such as sPD-L1 and sCTLA-4, there is a lack of discussion regarding assay standardization or pre-analytical variability, both of which are essential for clinical implementation.
Thank you for the constructive feedback and helpful comments.
Assay standardization and preanalytical variability are crucial for clinical use of biomarkers to ensure accurate and comparable results. Special attention is paid to ensure that the assays are robust to common confounding factors such as hemoglobin and bilirubin, which reduces preanalytical variability. However, detailed procedures such as sample storage are not fully controlled or may be a potential point of variability. Therefore, further studies are needed to establish universal standards for these biomarkers in clinical practice. [He, Y., Zhang, X., Zhu, M. et al. Soluble PD-L1: a potential dynamic predictive biomarker for immunotherapy in patients with proficient mismatch repair colorectal cancer. J Transl Med 21, 25 (2023). https://doi.org/10.1186/s12967-023-03879-0; Park, JJ., Thi, E.P., Carpio, V.H. et al. Checkpoint inhibition through small molecule-induced internalization of programmed death-ligand 1. Nat Commun 12, 1222 (2021).; Buchbinder EI, Desai A. CTLA-4 and PD-1 Pathways: Similarities, Differences, and Implications of Their Inhibition. Am J Clin Oncol. 2016 Feb;39(1):98-106. doi: 10.1097/COC.0000000000000239. PMID: 26558876; PMCID: PMC4892769].Key elements of standardization should include basic steps in the validation of assay methods, such as standard curve development, assay sensitivity, detection limits, quantification limits, assay repeatability, assay specificity, dilution linearity, and post-dose recovery. These elements ensure that assays are precise, reproducible, and suitable for clinical use, which is essential for comparability of results between laboratories.
Preanalytical variability is one of the parameters that influence the obtained results and biomarker parameter values. Preanalytical variability refers to variations that may occur before sample analysis, such as sample collection, storage, and processing. For samples, steps such as interference testing, precise sample type and sample handling, i.e., when biomarker measurement should be performed, whether before or after surgical interventions, are important, suggesting standardized sample collection procedures, although details such as storage conditions and processing time are not widely discussed. The influence of tumor type and surgical intervention type is also important. Studies indicate the need for harmonization of biomarker measurement methods across laboratories. For example, other studies often use ELISA to measure these biomarkers, but without standardized protocols, results may be incomparable between laboratories. The need for common reference standards and sample handling protocols is crucial for the clinical application of these biomarkers. The accompanying publication fills a gap in the review by providing detailed information on assay standardization for sPD-L1 and sCTLA-4, which is crucial for their clinical application. It also addresses preanalytical variability, indicating that the assays are robust to common interferences, although detailed preanalytical procedures require further discussion. Further studies are needed to establish universal standards, which is necessary for the broad application of these biomarkers in clinical practice.[ [Goto, M., Chamoto, K., Higuchi, K. et al. Analytical performance of a new automated chemiluminescent magnetic immunoassays for soluble PD-1, PD-L1, and CTLA-4 in human plasma. Sci Rep 9, 10144 (2019). https://doi.org/10.1038/s41598-019-46548-3]
- The review refers to levels of cytokines, such as IL-6, IL-8, IL-10, but does not consider either intra-patient temporal variation or the relationships between these cytokines and other immune parameters over the course of therapy.
Thank you for this suggestion.
Studies show that dynamic changes in cytokine profiles, such as decreased IL-6 and IL-8 levels, may reflect the short-term efficacy of immunotherapy. [Fisher DT, Appenheimer MM, Evans SS. The two faces of IL-6 in the tumor microenvironment. Semin Immunol. 2014 Feb;26(1):38-47. doi: 10.1016/j.smim.2014.01.008. Epub 2014 Mar 3. PMID: 24602448; PMCID: PMC3970580.] Changes in CCL11, IL1RA, and IL17A levels after treatment were associated with long-term progression-free survival. [Tyan K, Baginska J, Brainard M, Giobbie-Hurder A, Severgnini M, Manos M, Haq R, Buchbinder EI, Ott PA, Hodi FS, Rahma OE. Cytokine changes during immune-related adverse events and corticosteroid treatment in melanoma patients receiving immune checkpoint inhibitors. Cancer Immunol Immunother. 2021 Aug;70(8):2209-2221. doi: 10.1007/s00262-021-02855-1. Epub 2021 Jan 22. PMID: 33481042; PMCID: PMC10991353.]
In addition, interactions between cytokines (e.g. IL-10 inhibiting the production of IL-1, IL-6, IL-12) are complex and affect the overall immune response. [Liu J, Mao Y, Mao C, Wang D, Dong L, Zhu W. An On-Treatment Decreased Trend of Serum IL-6 and IL-8 as Predictive Markers Quickly Reflects Short-Term Efficacy of PD-1 Blockade Immunochemotherapy in Patients with Advanced Gastric Cancer. J Immunol Res. 2024 May 14;2024:3604935. doi: 10.1155/2024/3604935. PMID: 38774604; PMCID: PMC11108694.] Different checkpoint inhibitors, such as CTLA-4 and PD-1 blockade, may induce distinct immunologic changes and cytokine profiles. [Guo AJ, Deng QY, Dong P, Zhou L, Shi L. Biomarkers associated with immune-related adverse events induced by immune checkpoint inhibitors. World J Clin Oncol. 2024 Aug 24;15(8):1002-1020. doi: 10.5306/wjco.v15.i8.1002. PMID: 39193157; PMCID: PMC11346067.] Ignoring this temporal variability and interplay limits a full understanding of the role of cytokines as biomarkers. Single measurements at a single time point may not capture the complexity of the immune response and the dynamics of the tumor microenvironment. Dynamic monitoring of biomarkers is critical. Rather than relying on single measurements at a single time point, future studies should focus on profiling cytokines and other immune parameters in a serial manner to capture their kinetics and interplay during treatment. Such analysis can provide much more precise information about response to therapy, early detection of resistance and prediction of adverse events.
- While the current text quotes a meta-analysis linking low levels of sPD-L1 to superior outcomes, it falls short of providing cohort size, inclusion criteria, or statistical significance to qualify its findings.
We appreciate the reviewer’s observation. However, we would like to clarify that the current manuscript is a narrative review, not a systematic review or meta-analysis. As such, its primary aim is not to systematically extract, analyze, and present all individual study parameters such as cohort sizes, inclusion criteria, or precise statistical significance values for every cited study. Instead, our intention is to synthesize and contextualize current findings in a broader conceptual framework, highlighting key themes, biological mechanisms, and emerging trends relevant to the topic.
It is important to note that all clinical findings included in our review are based on statistically significant results reported in the original studies. To further illustrate this approach and to address potential concerns regarding data robustness, we have added an exemplary excerpt to the manuscript from a representative study demonstrating how such data are typically interpreted. The included paragraph reads:
"In a clinical study involving 33 patients with hepatocellular carcinoma (HCC) treated with nivolumab, a PD-1 inhibitor, high diastolic whole blood viscosity (WBV) (>16.0 cP) was associated with poorer overall survival (OS) and progression-free survival (PFS) compared to the group with lower WBV (≤16.0 cP). The differences were close to statistical significance (P=0.069 for OS, P=0.067 for PFS), which may be due to the small sample size. The objective response rate was 25.0% in the low WBV group (6/24, including 1 complete response and 5 partial responses) compared to 0% in the high WBV group (0/9), although the difference was not statistically significant (P=0.409). The study suggests that WBV may serve as a potential biomarker in HCC, particularly in the context of immunotherapy, but emphasizes the need for larger, prospective studies."
[Han JW, Sung PS, Jang JW, Choi JY, Yoon SK. Whole blood viscosity is associated with extrahepatic metastases and survival in patients with hepatocellular carcinoma. PLoS One. 2021 Dec 2;16(12):e0260311. doi: 10.1371/journal.pone.0260311. PMID: 34855786; PMCID: PMC8638904]
This example illustrates our commitment to transparency in referencing clinically relevant findings, while also respecting the conventions of narrative review methodology, which does not require exhaustive reporting of statistical data for every study referenced.
We hope this clarifies the rationale behind our approach and aligns with the expectations for a narrative review format.
- The addition of myeloid-derived suppressor cells (MDSCs) as a biomarker is beneficial but the review could have incorporated harmonized strategies for their detection and quantification which is a major hurdle to clinical application.
Thank you for this comment.
Studies indicate heterogeneity of MDSC populations and considerable interlaboratory variability in their phenotyping and quantification by flow cytometry, especially for granulocytic subsets. [ Mandruzzato S, Brandau S, Britten CM, Bronte V, Damuzzo V, Gouttefangeas C, Maurer D, Ottensmeier C, van der Burg SH, Welters MJ, Walter S. Toward harmonized phenotyping of human myeloid-derived suppressor cells by flow cytometry: results from an interim study. Cancer Immunol Immunother. 2016 Feb;65(2):161-9. doi: 10.1007/s00262-015-1782-5. Epub 2016 Jan 4. PMID: 26728481; PMCID: PMC4726716.]The lack of a uniform, universally accepted classification of human MDSCs (both in terms of subset types and identification markers) and differences in gating strategies are the main sources of this variability.[ Flores-Campos R, García-Domínguez DJ, Hontecillas-Prieto L, Jiménez-Cortegana C, de la Cruz-Merino L, Sánchez-Margalet V. Flow cytometry analysis of myeloid derived suppressor cells using 6 color labeling. Methods Cell Biol. 2024;190:1-10. doi: 10.1016/bs.mcb.2024.08.006. Epub 2024 Sep 28. PMID: 39515873.] This makes comparing study results and establishing reliable cutoff values ​​for clinical assessment extremely difficult. Further efforts to harmonize protocols and standardize methodologies are essential for MDSCs to become routinely useful biomarkers in oncology. [Gabrilovich DI. Myeloid-Derived Suppressor Cells. Cancer Immunol Res. 2017 Jan;5(1):3-8. doi: 10.1158/2326-6066.CIR-16-0297. PMID: 28052991; PMCID: PMC5426480.]
Solving the heterogeneity problem in MDSC quantification is crucial. The complexity of MDSC subsets and the lack of consensus on the best combinations of surface markers for their identification by flow cytometry pose significant challenges. Standardization of gating strategies and protocols is essential to obtain consistent and comparable data, enabling their reliable clinical application.
- The review does not consider that ctDNA, while the most studied blood biomarker/marker, may not always adequately reflect tumor heterogeneity or clonal evolution, factors that similarly affect detection and interpretation over time.
Thank you for this suggestion.
Circulating tumor DNA (ctDNA) is a valuable tool for monitoring tumor burden and tumor dynamics in real time. However, ctDNA, although reflecting genomic alterations, may not always fully capture the complex spatial heterogeneity (differences between tumor regions or metastases) or clonal evolution (emergence of new mutations or subclones over time) of the tumor [Moon GY, Dalkiran B, Park HS, Shin D, Son C, Choi JH, Bang S, Lee H, Doh I, Kim DH, Jeong WJ, Bu J. Dual Biomarker Strategies for Liquid Biopsy: Integrating Circulating Tumor Cells and Circulating Tumor DNA for Enhanced Tumor Monitoring. Biosensors (Basel). 2025 Jan 28;15(2):74. doi: 10.3390/bios15020074. PMID: 39996976; PMCID: PMC11852634]. ctDNA levels may be low in early stages of disease or in low-burden tumors, which limits its sensitivity. Furthermore, the ctDNA fraction may be variable and may not always reflect all tumor clones, especially those with low prevalence. To overcome these limitations, it is necessary to combine ctDNA analyses with other methods, such as mononuclear cell immunophenotyping, extracellular vesicle analysis, and plasma proteomics, as well as integrated approaches with circulating tumor cells (CTCs) [Monick S, Rosenthal A. Circulating Tumor DNA as a Complementary Prognostic Biomarker during CAR-T Therapy in B-Cell Non-Hodgkin Lymphomas. Cancers (Basel). 2024 May 15;16(10):1881. doi: 10.3390/cancers16101881. PMID: 38791959; PMCID: PMC11120115.; Nakamura Y, Shitara K. Development of circulating tumor DNA analysis for gastrointestinal cancers. ESMO Open. 2020 Jan;5(Suppl 1):e000600. doi: 10.1136/esmoopen-2019-000600. PMID: 32830648; PMCID: PMC7003376.].
Although ctDNA provides valuable real-time information on tumor dynamics, its limitations in fully reflecting tumor heterogeneity and clonal evolution emphasize the need for multimodal approaches. Combining ctDNA with other biomarkers and advanced analytical methods may provide a more comprehensive view of tumor biology and treatment response [Gao J, Wang H, Zang W, Li B, Rao G, Li L, Yu Y, Li Z, Dong B, Lu Z, Jiang Z, Shen L. Circulating tumor DNA functions as an alternative for tissue to overcome tumor heterogeneity in advanced gastric cancer. Cancer Sci. 2017 Sep;108(9):1881-1887. doi: 10.1111/cas.13314. Epub 2017 Jul 29. PMID: 28677165; PMCID: PMC5581520.]
- LDH is inexpensive and widely available as a prognostic marker as mentioned in this review but it has not been compared for its sensitivity or specificity against other systemic inflammation markers.
Thank you for the constructive comments.
For LDH and CRP, ROC-AUC analysis in melanoma patients showed that CRP (AUC=0.933) was superior to LDH (AUC=0.491) in discriminating stage IV melanoma. In the context of diagnosing stage IV melanoma, at a cutoff of 3.0 mg/l, CRP achieved a sensitivity of 0.769 and a specificity of 0.904, whereas LDH did not provide additional information compared with CRP. [Deichmann M, Kahle B, Moser K, Wacker J, Wüst K. Diagnosing melanoma patients entering American Joint Committee on Cancer stage IV, C-reactive protein in serum is superior to lactate dehydrogenase. Br J Cancer. 2004 Aug 16;91(4):699-702. doi: 10.1038/sj.bjc.6602043. PMID: 15280926; PMCID: PMC2364774.]
In non-small cell lung cancer (NSCLC), a study showed that both serum CRP and LDH levels, as well as their combination, could predict the response to checkpoint inhibitors. High LDH levels and low CRP levels were associated with unfavorable progression-free survival [Wei Y, Xu J, Huang X, Xie S, Lin P, Wang C, Guo Y, Zou S, Zhao Z, Wen W, Song Y, Bao Z, Zhang L, Liu W, Kong W, Wang W, He B, Zhang S, Zhou C, Chen Y, Yu Z. C-reactive protein and lactate dehydrogenase serum levels potentially predict the response to checkpoint inhibitors in patients with advanced non-small cell lung cancer. J Thorac Dis. 2023 Apr 28;15(4):1892-1900. doi: 10.21037/jtd-23-240. Epub 2023 Apr 17. PMID: 37197527; PMCID: PMC10183533]. In acute pancreatitis, LDH showed higher sensitivity (94.9%) but lower specificity (88.2%) compared to CRP (sensitivity 59.0%, specificity 97.4%) [Yin, X., Xu, J., Zhang, Q., Yang, L., & Duan, Y. (2020). Quantification analysis of lactate dehydrogenase and C-reactive protein in evaluation of the severity and prognosis of the acute pancreatitis. Cellular and Molecular Biology, 66(1), 122–125. https://doi.org/10.14715/cmb/2019.66.1.20].
In lung cancer, although the combination of CRP and neuron-specific enolase (NSE) showed high specificity (94%) and accuracy (82.67%), LDH levels were not statistically different in the patient groups [Bhatia, Chahat & Jaswal, Shivani & Sodhi, Mandeep & Kaur, Jasbinder & Aggarwal, Phiza & Aggarwal, Deepak & Saini, Varinder & Kaur, Ravinder. (2023). Evaluation of a Panel of Biomarkers in the Diagnosis of Lung Cancer: An Observational Study. Indian Journal of Respiratory Care. 12. 244-247. 10.5005/jp-journals-11010-1061.].
Analysis of the results obtained from the studies shows that CRP appears to be superior to LDH in staging melanoma, whereas their individual diagnostic powers differ in other conditions, such as acute pancreatitis. Importantly, studies in NSCLC and prostate cancer show that their combination can increase predictive power. This suggests that rather than one marker being universally "better," they may offer complementary information, with their optimal utility being dictated by the specific tumor type, stage, and clinical question.
- While there are calls for multi-biomarker combinations to increase predictive power, this review does not provide a framework or risk-scoring system for integrating the different biomarkers into a single risk stratification system.
Thank you for the constructive feedback.
The biomarkers might be used as a comprehensive, multi-parametric framework designed to stratify cancer patients based on their immune and tumor-related biomarker profiles. This system might integrate molecular, cellular, and inflammatory biomarkers to assess the immune competence and tumor aggressiveness of individual patients, enabling more precise prognostic predictions and informing therapeutic strategies.
At the core of the framework are eight key biomarkers that reflect various aspects of tumor biology and host immune response: circulating tumor DNA (ctDNA), lactate dehydrogenase (LDH), C-reactive protein (CRP), cytokine signaling (IL-6, IL-10, TNF-α), eosinophil count, regulatory T cells (Tregs), myeloid-derived suppressor cells (MDSCs), and monocyte levels. Each of these biomarkers might be assigned as favorable or unfavorable based on established thresholds in the literature.
Each biomarker contributes incrementally to the patient’s risk score, which is then used to classify individuals into risk categories. The risk score can be used both at baseline—to guide first-line treatment decisions—and longitudinally, to track changes in biomarker profiles in response to therapy. Its integrative approach allows clinicians to move beyond single-marker assessments, providing a holistic view of tumor-immune dynamics.
Moding EJ, Liu Y, Nabet BY, et al. Circulating Tumor DNA Dynamics Predict Benefit from Consolidation Immunotherapy in Locally Advanced Non-Small Cell Lung Cancer. Nat Cancer. 2020;1(2):176-183. doi:10.1038/s43018-019-0011-0
Diem S, Kasenda B, Spain L, et al. Serum lactate dehydrogenase as an early marker for outcome in patients treated with anti-PD-1 therapy in metastatic melanoma. Br J Cancer. 2016;114(3):256-261. doi:10.1038/bjc.2015.467
Xu Y, Ma K, Zhang F, et al. Association between baseline C‑reactive protein level and survival outcomes for cancer patients treated with immunotherapy: A meta‑analysis. Exp Ther Med. 2023;26(2):361. Published 2023 Jun 8. doi:10.3892/etm.2023.12060
Lee S, Margolin K. Cytokines in cancer immunotherapy. Cancers (Basel). 2011;3(4):3856-3893. Published 2011 Oct 13. doi:10.3390/cancers3043856
Tanaka A, Sakaguchi S. Regulatory T cells in cancer immunotherapy. Cell Res. 2017;27(1):109-118. doi:10.1038/cr.2016.151
Gabrilovich DI, Nagaraj S. Myeloid-derived suppressor cells as regulators of the immune system. Nat Rev Immunol. 2009;9(3):162-174. doi:10.1038/nri2506
Boutilier AJ, Elsawa SF. Macrophage Polarization States in the Tumor Microenvironment. Int J Mol Sci. 2021;22(13):6995. Published 2021 Jun 29. doi:10.3390/ijms22136995
- The review has references, highlighting the cost effectiveness of the NLR, PLR and LMR, but no comparative performance metrics or ROC analyses are provided to show that these are better than a more sophisticated (but costly) test.
We are truly thankful for the reviewer’s constructive and thoughtful remarks.
The diagnostic and prognostic values of MLR, NLR, PLR, CEA, and CA19-9 were assessed using receiver operating characteristic (ROC) curve analysis and the chi-square test in a retrospective analysis was conducted involving 783 patients diagnosed with colorectal cancer (CRC) and 1,232 age-matched healthy controls. Levels of MLR, NLR, and PLR were significantly elevated in CRC patients compared to controls. The areas under the ROC curve (AUC) for MLR, CEA, PLR, NLR, and CA19-9 were 0.739, 0.726, 0.683, 0.610, and 0.603, respectively. The combination of CEA and MLR yielded the highest diagnostic performance (AUC = 0.815), outperforming other combinations, including CEA with CA19-9. These findings indicate that MLR is a superior diagnostic marker for colorectal cancer compared to NLR and PLR, whereas NLR may hold greater prognostic value for CRC patients [Kang, Yanli, et al. "Compare the Diagnostic and Prognostic Value of MLR, NLR and PLR in CRC Patients." Clinical laboratory 9 (2021).].
Another study highlighting the superiority of neutrophil-to-lymphocyte ratio (NLR) and platelet-to-lymphocyte ratio (PLR) over more costly diagnostic methods focused on gastric cancer. Most patients with gastric cancer remain asymptomatic until the disease reaches advanced stages. Although gastroscopy remains the gold standard diagnostic tool recommended by clinical guidelines, its invasive nature and high cost limit its widespread use in screening programs for early detection. Commonly used tumor markers, such as carcinoembryonic antigen (CEA) and carbohydrate antigen 19-9 (CA19-9), exhibit limited sensitivity and specificity, thereby restricting their diagnostic utility in gastric cancer. This retrospective study included 2,606 patients diagnosed with gastric cancer (GC) and 3,219 healthy controls, monitored over a three-year period. Peripheral blood samples were analyzed to assess levels of NLR, PLR, CEA, and CA19-9. Optimal cutoff values for NLR and PLR were established through receiver operating characteristic (ROC) curve analysis.
The findings demonstrated that systemic inflammatory markers NLR and PLR were significantly associated with gastric cancer diagnosis, particularly among male patients. These results suggest that assessment of inflammatory markers in peripheral blood may serve as a valuable tool for identifying high-risk populations for gastric cancer.[ Fang, Tianyi, et al. "Diagnostic sensitivity of NLR and PLR in early diagnosis of gastric cancer." Journal of immunology research 2020.1 (2020): 9146042]
However, without further clinical trial data supporting the claim that rheological parameters can become a new standard in the monitoring of immunotherapy, this statement seems speculative.
Thank you for this insightful and accurate comment .
Already in 2008 in the work: Késmárky G, Kenyeres P, Rábai M, Tóth K. Plasma viscosity: a forgotten variable. Clin Hemorheol Microcirc. 2008;39(1–4):243–6; the authors pointed out that plasma viscosity is a forgotten parameter, and in turn the author of the work: Kwaan HC. Role of plasma proteins in whole blood viscosity: a brief clinical review. Clin Hemorheol Microcirc. 2010;44(3):167–76; in his review presented the influence of plasma proteins on whole blood viscosity. Figure 3 presents the determinants of whole blood viscosity. Based on the review, the Rheological Parameters section presents the relationship between the patient's condition and the values ​​of hematocrit, whole blood viscosity, plasma viscosity or aggregability and deformability of erythrocytes. In a review paper by one of the co-authors (item 49), it was shown that one of the concepts of cancer treatment may be modification of excessive viscosity, especially of plasma, which may improve the body's response to radio- and chemotherapy and prevent thrombosis, which is a common complication in neoplastic diseases. It is also worth noting that analysis of changes in the values ​​of hemorheological parameters, such as aggregability of erythrocytes and plasma viscosity, may be significant, also because thromboembolic events may be the cause of complications in the process of surgical and chemotherapeutic treatment of cancer. On this basis, the authors believe that the statement that further research is needed seems justified. Studies of the values ​​of hemorheological parameters for monitoring immunotherapy could be justified. In line 227, a reference to figure 3 was added.
- Therefore, due to lack of statistical meta-analyses or pooled hazard ratios to substantiate the prognostic significance of the biomarkers addressed in this systematic review, the strength of the clinical recommendations mentioned will be limited.
Thank you for this insightful and accurate comment. We agree that the lack of statistical meta-analyses and pooled hazard ratios limits the ability to definitively establish the prognostic value of the biomarkers discussed and consequently affects the strength of the clinical recommendations presented.
However, we would like to emphasize that our paper is a literature review, not a systematic review or meta-analysis. The aim of this review was to provide a comprehensive overview of the currently available evidence and to identify potentially clinically relevant biomarkers that may gain greater significance after further validation in well-designed prospective studies and pooled analyses. We have also pointed out the knowledge gaps that need to be addressed to formulate stronger recommendations. These limitations have been clearly acknowledged in the conclusion sections of the manuscript.
- That aside, the review points out that blood-based markers are easier for both patients and clinicians to access in comparison but fails to mention differences in factors such as availability of assays, calibration or interpretation in clinical laboratories.
The following paragraphs were added to provide more details on these differences:
While tissue-based biomarkers, such as PD-L1 expression, TMB and MSI, are currently most often used biomarkers for ICIs treatment, they require obtaining a tissue sample from the tumor. Immunohistochemical (IHC) assays for assessment of PD-L1 expression in tumor and immune cells were developed along PD-(L)1 inhibitors. Standardized assays include 22C3 and 28-8 pharmDx on Dako platforms, as well as SP142 and SP263 on Ventana platforms. The results of PD-L1 testing may vary depending on the monoclonal antibody, IHC platform and interpreting pathologist. Studies showed that 28-8, 22C3 and SP263 assays were highly concordant for tumor cells staining, but not for immune cells staining [Büttner R, Gosney JR, Skov BG, et al. Programmed Death-Ligand 1 Immunohistochemistry Testing: A Review of Analytical Assays and Clinical Implementation in Non-Small-Cell Lung Cancer. J Clin Oncol. 2017;35(34):3867-3876. doi:10.1200/JCO.2017.74.7642]. A multicenter study by Adam et al. analized concordance of aforementioned standardized assays as well as 27 laboratory-developed tests, with combinations of five anti-PD-L1 monoclonal antibodies (28-8, 22C3, E1L3N, SP142 and SP263) and three types of IHC platforms (Dako Autostainer Link 48, Leica Bond and Ventana BenchMark Ultra). 14 of laboratory-developed tests (51,8%) demonstrated similar concordance to the standardized assays [Adam J, Le Stang N, Rouquette I, et al. Multicenter harmonization study for PD-L1 IHC testing in non-small-cell lung cancer. Ann Oncol. 2018;29(4):953-958. doi:10.1093/annonc/mdy014].
Blood-based biomarkers have the advantage of being obtained with a simple blood draw instead of a biopsy. However, the diagnostic laboratory needs access to necessary equipment and qualified personnel. The cost and availability depends on the specific biomarker. NLR, LMR and PLR can be calculated at any laboratory with a hematology analyzer as elements of routine complete blood count. Variability can be minimized by quality control and analyzing fresh blood samples [Kim MS, Park CJ, Namgoong S, Kim SI, Cho YU, Jang S. Effective and Practical Complete Blood Count Delta Check Method and Criteria for the Quality Control of Automated Hematology Analyzers. Ann Lab Med. 2023;43(5):418-424. doi:10.3343/alm.2023.43.5.418; Kim MS, Park CJ, Namgoong S, Kim SI, Cho YU, Jang S. Effective and Practical Complete Blood Count Delta Check Method and Criteria for the Quality Control of Automated Hematology Analyzers. Ann Lab Med. 2023;43(5):418-424. doi:10.3343/alm.2023.43.5.418]. If necessary, samples can be sent to a reference laboratory, but optimal storage conditions should be provided [Åstrand A, Wingren C, Walton C, et al. A comparative study of blood cell count in four automated hematology analyzers: An evaluation of the impact of preanalytical factors. PLoS One. 2024;19(5):e0301845. Published 2024 May 24. doi:10.1371/journal.pone.0301845; Oliveira LR, Simionatto M, Cruz BR, et al. Stability of complete blood count in different storage conditions using the ABX PENTRA 60 analyzer. Int J Lab Hematol. 2018;40(3):359-365. doi:10.1111/ijlh.12797]. Some rheological parameters of blood (such as hematocrit) can be assessed as part of the complete blood count, but analyzing erythrocyte deformability and whole blood viscosity requires additional equipment [Rosencranz R, Bogen SA. Clinical laboratory measurement of serum, plasma, and blood viscosity. Am J Clin Pathol. 2006;125 Suppl:S78-S86. doi:10.1309/FFF7U8RRPK26VAPY].
- The ability of immunotherapy-induced immune-related adverse events (irAEs) to confound interpretation of specific systemic biomarkers is not addressed, either.
Thank you for this comment.
The authors did not find any publications from the past five years that addressed the potential of immunotherapy-induced immune-related adverse events (irAEs) to confound the interpretation of specific systemic biomarkers. This potential influence certainly warrants further investigation. Existing studies on irAEs have mainly focused on the opposite relationship – identifying risk factors and predictive biomarkers for the occurrence of irAEs, including blood-based biomarkers.
- Chennamadhavuni A, Abushahin L, Jin N, Presley CJ, Manne A. Risk Factors and Biomarkers for Immune-Related Adverse Events: A Practical Guide to Identifying High-Risk Patients and Rechallenging Immune Checkpoint Inhibitors. Front Immunol. 2022;13:779691. Published 2022 Apr 26. doi:10.3389/fimmu.2022.779691
- Berner F, Flatz L. Autoimmunity in immune checkpoint inhibitor-induced immune-related adverse events: A focus on autoimmune skin toxicity and pneumonitis. Immunol Rev. 2023; 318: 37-50. doi:10.1111/imr.13258.
- Michailidou D, Khaki AR, Morelli MP, Diamantopoulos L, Singh N, Grivas P. Association of blood biomarkers and autoimmunity with immune related adverse events in patients with cancer treated with immune checkpoint inhibitors. Sci Rep. 2021;11(1):9029. Published 2021 Apr 27. doi:10.1038/s41598-021-88307-3
- Raynes G, Stares M, Low S, et al. Immune-Related Adverse Events, Biomarkers of Systemic Inflammation, and Survival Outcomes in Patients Receiving Pembrolizumab for Non-Small-Cell Lung Cancer. Cancers (Basel). 2023;15(23):5502. Published 2023 Nov 21. doi:10.3390/cancers15235502
- Wang J, Ma Y, Lin H, Wang J, Cao B. Predictive biomarkers for immune-related adverse events in cancer patients treated with immune-checkpoint inhibitors. BMC Immunol. 2024;25(1):8. Published 2024 Jan 24. doi:10.1186/s12865-024-00599-y
- Johnson DB, Balko JM. Biomarkers for Immunotherapy Toxicity: Are Cytokines the Answer? Clin Cancer Res. 2019 Mar 1;25(5):1452-1454. doi: 10.1158/1078-0432.CCR-18-3858. Epub 2018 Dec 26. PMID: 30587548; PMCID: PMC6397678.
- Guo AJ, Deng QY, Dong P, Zhou L, Shi L. Biomarkers associated with immune-related adverse events induced by immune checkpoint inhibitors. World J Clin Oncol. 2024;15(8):1002-1020. doi:10.5306/wjco.v15.i8.1002
- Although it references historical studies on the micro-circulation and viscosity of tumors, no recent imaging or computational modeling studies that quantify those observations appear to be available.
Thank you for that suggestion.
The authors did not find any papers published in the last five years that discussed aspects related to the viscosity of whole blood in the presence of a developing tumor or other neoplastic lesion. Contemporary papers on microcirculation in the tumor refer to the characteristics of the vessels, not blood parameters. However, papers from the turn of the 20th and 21st centuries provided information that the development of cancer may change the rheological parameters of the blood, which was described by one of the co-authors in a 2019 paper (item 49). The analysis of the hemorheological profile of patients with cancer presented in the paper indicated increased values ​​of parameters such as aggregability of erythrocytes, erythrocyte stiffness (reduced deformability), plasma viscosity and increased levels of fibrinogen and globulins (proteins affecting plasma viscosity, and thus the viscosity of whole blood) compared to the group of healthy individuals. In the works from that period, it was pointed out that in patients with advanced disease, an increase in whole blood viscosity was also observed with normal or reduced hematocrit levels. The authors of the works from the 1990s pointed out that these changes disturbed blood flow in the microcirculation, but also affected the microcirculation of the tumor, and thus the onset of metastases and the effectiveness of treatment. The aim of the work submitted for publication was not to analyze the microcirculation in the tumor, and therefore the authors did not develop this aspect of the work.
- These studies do not report the impact of baseline comorbidities (e.g. CV disease, diabetes) on blood-based biomarker levels, which could alter their population specificity in the cancer context.
We appreciate the reviewer’s valuable comment.
This paper did not analyze the influence of the occurrence of comorbidities on the levels of biomarkers in the blood. There are many papers that present analyses of the relationships between the values ​​of hemorheological parameters in various circulatory system diseases or diabetes. The authors thank the reviewers for this comment. Such a review could provide the basis for a new paper. In this presented study, the authors did not conduct a review to analyze the development of neoplastic disease depending on the occurrence of comorbidity. Many sources only drew attention to the development of neoplastic disease and to the fact that increased whole blood viscosity may affect the incidence of venous thromboembolism, but there was also information that in patients with hyperviscosity, symptoms may be confused with stroke, dyspnea may be confused with pulmonary embolism or heart failure, or an altered mental state may be confused with sepsis.
- Although insights into the temporal dynamics of these biomarkers are critical to improving their clinical utility (e.g., during PD-1 vs. CTLA-4 blockade) they are not explicitly addressed.
We appreciate the reviewer’s valuable comment.
Temporal dynamics refer to the evolution of biomarkers, such as immune cell populations, gene expression profiles, or circulating tumor DNA (ctDNA), during immunotherapy. Unlike static pre-treatment measurements, longitudinal assessments capture the immune system’s response to treatment, providing insights into mechanisms of action and clinical outcomes. These dynamics are particularly relevant for PD-1 and CTLA-4 inhibitors, which modulate T-cell activity at different stagesPD-1 primarily in peripheral tissues later in the immune response, and CTLA-4 in lymph nodes early on. [Buchbinder EI, Desai A. CTLA-4 and PD-1 Pathways: Similarities, Differences, and Implications of Their Inhibition. Am J Clin Oncol. 2016;39(1):98-106. doi:10.1097/COC.0000000000000239]
PD-1 and CTLA-4 inhibitors target distinct immune checkpoints, leading to different temporal biomarker profiles. PD-1 blockade primarily enhances exhausted CD8+ T-cell proliferation early in treatment, as seen in lung cancer patients where PD-1+CD8+ T cells peaked within the first week [Kamphorst AO, Pillai RN, Yang S, et al. Proliferation of PD-1+ CD8 T cells in peripheral blood after PD-1-targeted therapy in lung cancer patients. Proc Natl Acad Sci U S A. 2017;114(19):4993-4998. doi:10.1073/pnas.1705327114]. Conversely, CTLA-4 blockade promotes CD4+ Th1-like effector cell expansion and Treg depletion, with effects more evident at later time points (e.g., 3 weeks post-treatment) [Wei SC, Levine JH, Cogdill AP, et al. Distinct Cellular Mechanisms Underlie Anti-CTLA-4 and Anti-PD-1 Checkpoint Blockade. Cell. 2017;170(6):1120-1133.e17. doi:10.1016/j.cell.2017.07.024]. Combination therapy synergizes these effects, sustaining robust T-cell responses, as evidenced by increased Ki-67+ CD8+ T cells in peripheral blood post-ipilimumab plus nivolumab [Wei SC, Anang NAS, Sharma R, et al. Combination anti-CTLA-4 plus anti-PD-1 checkpoint blockade utilizes cellular mechanisms partially distinct from monotherapies. Proc Natl Acad Sci U S A. 2019;116(45):22699-22709. doi:10.1073/pnas.1821218116].
Implementing longitudinal biomarker analysis faces challenges, including the invasiveness of serial biopsies and the need for standardized timing protocols. Optimal timing for assessment varies by biomarker and treatment; for instance, PD-L1 IHC requires staining within 6 months of tissue sectioning, and T-cell assays need prompt PBMC isolation to avoid suppression [Masucci GV, Cesano A, Hawtin R, et al. Validation of biomarkers to predict response to immunotherapy in cancer: Volume I - pre-analytical and analytical validation. J Immunother Cancer. 2016;4:76. Published 2016 Nov 15. doi:10.1186/s40425-016-0178-1]. Advances in liquid biopsies and high-throughput technologies, like mass cytometry and single-cell sequencing, could facilitate non-invasive, frequent monitoring. Future research should focus on validating optimal time points (e.g., 1, 3, 6 weeks) and developing combination biomarker strategies to enhance predictive accuracy.
The temporal dynamics of biomarkers during PD-1 and CTLA-4 blockade provide critical insights into immune responses, enabling improved patient stratification, response monitoring, and resistance management. Longitudinal studies demonstrate distinct patterns between PD-1 and CTLA-4 therapies, with combination approaches amplifying these effects. By integrating serial biomarker assessments into clinical practice, immunotherapy can be optimized, paving the way for precision oncology.
- Although the review appropriately notes that most of the biomarkers discussed are unvalidated in large-scale studies, it does not set out specific appropriate trial designs or biomarker-driven endpoints that could direct future research
We appreciate the reviewer’s valuable comment. In response, we have added a dedicated paragraph to the manuscript discussing how future research should be structured to validate the clinical utility of NLR, LMR, PLR, and rheological parameters.
Specifically, we now state that future prospective and randomized clinical trials should be designed to assess dynamic changes in these biomarkers during immunotherapy. We emphasize the importance of including biomarker-driven endpoints such as progression-free survival (PFS), overall survival (OS), objective response rate (ORR), and incidence of immune-related adverse events (irAEs). Additionally, we propose stratification based on baseline values and their changes to help define cut-off thresholds and facilitate the integration of these cost-effective biomarkers into predictive models. We also highlight the potential of longitudinal monitoring to identify early indicators of therapeutic response or resistance.
ostatni akapit w pracy:
In future research, prospective and randomized clinical trials should be designed to assess the dynamic changes in NLR, LMR, PLR, and rheological parameters over the course of immunotherapy. These studies should include biomarker-driven endpoints such as progression-free survival (PFS), overall survival (OS), objective response rate (ORR), and incidence of immune-related adverse events (irAEs). Stratification based on baseline values and their changes could help to define cut-off thresholds and integrate these cost-effective biomarkers into predictive models. Moreover, longitudinal monitoring of these parameters may uncover their utility as early indicators of therapeutic response or resistance.

Reviewer 2 Report
Comments and Suggestions for Authors
1-According to my knowledge, cell-free DNA (cfDNA) is released by normal blood cells, primary lesions and metastases, and circulating tumor cells. The cfDNA levels at baseline and during treatment might be good biomarkers. circulating ncRNA in body fluids could serve as a very valuable non-invasive biomarker for patient selection and monitoring. Adding about these biomarkers would improve the review level.
2- lane 193, the abbreviation of platelet-to-lymphocyte ratio, PLR, can only be used because it has been already specified on lane 185.
3- Define ‘’ MPV’’ in the text, lane 275.
Author Response
Dear Reviewer,
We sincerely thank you for their valuable comments and suggestions, which have helped us improve the clarity and quality of our manuscript. All responses to the reviewer’s remarks have been addressed in the revised text and are highlighted in a different color for ease of reference.
- According to my knowledge, cell-free DNA (cfDNA) is released by normal blood cells, primary lesions and metastases, and circulating tumor cells. The cfDNA levels at baseline and during treatment might be good biomarkers. circulating ncRNA in body fluids could serve as a very valuable non-invasive biomarker for patient selection and monitoring. Adding about these biomarkers would improve the review level.
Thank you for your valuable comment. We have now added a relevant fragment in the Discussion section, highlighting (green color also) the potential of circulating non-coding RNAs (ncRNAs) in body fluids as promising non-invasive biomarkers for patient stratification and monitoring.
- 2-lane 193, the abbreviation of platelet-to-lymphocyte ratio, PLR, can only be used because it has been already specified on lane 185.
Thank you for pointing this out. We have removed the redundant repetition of the abbreviation PLR, as it was already introduced earlier in the text.
- 3- Define ‘’ MPV’’ in the text, lane 275.
Thank you for your observation. We have now defined the abbreviation "MPV" (mean platelet volume) at its first mention in the text to ensure clarity for the reader.
Reviewer 3 Report
Comments and Suggestions for Authors
This review describe very well the role of blood-based biomarkers as predictive and prognostic factors in patients with solid tumors treated with immunotherapy, and provides a good overview and perspective on how to use them in the clinic.
There are two things that need to be included in the review to make it stronger and more complete
1) Table 2: authors included studies depicting the findings of immunotherapy used as single agent. The table should also include clinical trials that used a combination of 2 or more immunotherapies and authors should add a new column describing the phase of clinical trial for each study reported in the table.
2) Authors should include a table 3 describing the role of ctDNA, LDH, CRP, Cytokine signaling, eosinophiles, Tregs, MDSCs and monocytes in affecting the efficacy of immunotherapy (single agent or combination of 2 or more agents) in solid tumors observed in clinical trials. Table 3 should resemble the template suggested for table 2
Author Response
Dear Reviewer,
Thank you very much for your insightful and valuable feedback, which has greatly contributed to improving our manuscript. We have carefully considered your suggestions and made the following updates:
- Table 2: authors included studies depicting the findings of immunotherapy used as single agent. The table should also include clinical trials that used a combination of 2 or more immunotherapies and authors should add a new column describing the phase of clinical trial for each study reported in the table.
In response to your first comment, we have revised Table 2 (Table 3 – in revised version of our manuscript) to include the type and phase of each clinical trial listed. This addition provides a clearer context for the studies involving single-agent immunotherapy.
- Authors should include a table 3 describing the role of ctDNA, LDH, CRP, Cytokine signaling, eosinophiles, Tregs, MDSCs and monocytes in affecting the efficacy of immunotherapy (single agent or combination of 2 or more agents) in solid tumors observed in clinical trials. Table 3 should resemble the template suggested for table 2
Regarding your suggestion to include clinical trials combining two or more immunotherapies and their associated biomarker responses, as well as the proposal for a new Table describing the role of ctDNA, LDH, CRP, cytokine signaling, eosinophils, Tregs, MDSCs, and monocytes in immunotherapy efficacy, we believe these are highly relevant and important topics. However, given the depth and complexity of these subjects, we plan to address them comprehensively in a separate review article to ensure they receive the thorough analysis they deserve.
We greatly appreciate your thoughtful comments, which have enhanced the clarity and quality of our work.
Reviewer 4 Report
Comments and Suggestions for Authors
It is an interesting and well-written article.
As indicated by the authors, the most assessed peripheral blood biomarker remains ctDNA. The cfDNA is relatively new and well-studied biomarker. However, the text content concerning cfDNA is small. It would be better if the authors can increase the text content of cfDNA as the biomarker in solid tumors.
What is the main question addressed by the research?
It is a review article, which discusses critical pathways involved in cancer development, systemic inflammatory response markers (neutrophil to lymphocyte ratio (NLR), lymphocyte to monocyte ratio (LMR), platelet to lymphocyte ratio (PLR)) and rheological parameters as prognostic and predictive factors in immunotherapy.
Do you consider the topic original or relevant to the field? Does it address a specific gap in the field? Please also explain why this is/ is not the case.
The topic is interesting because it provides new elements in the field. It is also a timely article because many clinical doctors look for the information.
What does it add to the subject area compared with other published material?
This article deals with some biomarkers analyzed in clinical laboratory with establisked techniques as well as the emerging biomaker such as cfDNA.
What specific improvements should the authors consider regarding the methodology?
It is a review article without problem concerning the methodology. However, the text content of cfDNA needs to be increased.
Are the conclusions consistent with the evidence and arguments presented and do they address the main question posed? Please also explain why this is/is not the case.
The conclusion are appropriate for a review article.
Are the references appropriate?
The references are large and adequate.
Any additional comments on the tables and figures.
The quality of figures is high, and the quality of tables is acceptable.
Author Response
Dear Reviewer,
We sincerely thank you for their valuable comments and suggestions, which have helped us improve the clarity and quality of our manuscript.
We sincerely thank the Reviewer for the positive and valuable feedback. We appreciate your comment regarding the cfDNA content. This topic will be addressed in more detail in our upcoming studies focused on advanced biomarkers for monitoring immunotherapy in cancer patients, where we will also include a discussion on epigenetic markers.
Reviewer 5 Report
Comments and Suggestions for Authors
This review discusses the use of blood-based biomarkers as predictive and prognostic factors in patients with solid tumors undergoing immunotherapy. The following points offer suggestions for potential improvement:
- In the Graphical Abstract, the organizational structure and categorization of biomarkers in the graphical abstract could be clarified for accuracy. Additionally, several brackets appear to be missing, which affects readability.
- In the Introduction section, expanding the introduction to include more background on why specific cell counts/ratios and rheological parameters are being investigated as potential biomarkers in the context of immunotherapy would better contextualize the detailed sections that follow.
- Figure 1 contains several incorrect labels or typographical errors that impede its ability to convey helpful information; a redesign or thorough correction is recommended.
- Some concepts or roles of biomarkers (e.g., LDH as an indicator of cell damage/tumor burden, or MDSCs' immunosuppressive role ) are introduced, then elaborated upon in the "Molecular Mechanisms" section, and sometimes reiterated in the "Discussion." For a "Mini Review," streamlining these explanations by perhaps integrating the primary mechanism when the biomarker is first introduced could enhance conciseness and reduce repetition.
- A more appropriate placement of Table 1 and Table 2 could be considered to ensure they are introduced in a context that optimally supports the surrounding text and the overall flow of the review.
- The text notes that changes in hemorheological parameters are observed in many diseases. When proposing them as biomarkers for immunotherapy response, a key challenge is dissecting changes specific to the tumor-immunotherapy interaction versus systemic changes due to general inflammation, or cancer progression itself, independent of immunotherapy efficacy. This point could be highlighted more.
- The discussion on hematocrit value in hematological neoplasms and the mention of COVID-19 vaccinations in multiple myeloma patients are quite specific to hematological contexts. While broadly related to blood properties, direct relevance as biomarkers for immunotherapy response in solid tumors is less clear and could potentially dilute the focus of this section for a mini review.
- The review rightly states that "the data for solid tumors affecting rheological blood properties is firmly limited." While the potential is highlighted, the specific mechanisms by which immunotherapy in solid tumors alter these parameters to predict response needs careful exploration.
- While Figure 4 effectively illustrates the diverse array of blood-based markers alongside FDA-approved tumor-based markers, providing a speculative or evidence-based example of how these different types of markers might be integrated into a multi-parametric predictive model would greatly enhance this section.
- The overall structure of the manuscript is not clear, particularly concerning Section 3. For instance, a significant portion of the content under Section 3, titled "Inflammatory Response Markers – NLR, LMR, PLR," does not directly pertain to these specific markers, which is really confusing.
Author Response
Dear Reviewer,
We sincerely thank you for their valuable comments and suggestions, which have helped us improve the clarity and quality of our manuscript. All responses to the reviewer’s remarks have been addressed in the revised text and are highlighted in a different color for ease of reference.
- In the Graphical Abstract, the organizational structure and categorization of biomarkers in the graphical abstract could be clarified for accuracy. Additionally, several brackets appear to be missing, which affects readability.
Thank you for your valuable feedback. We have revised the Graphical Abstract to improve the organizational structure and clarify the categorization of biomarkers. We ensured that the visual flow more accurately reflects the relationships between key pathways and components. Additionally, we reviewed the figure for formatting issues and have added the missing brackets to enhance readability and precision.
- In the Introduction section, expanding the introduction to include more background on why specific cell counts/ratios and rheological parameters are being investigated as potential biomarkers in the context of immunotherapy would better contextualize the detailed sections that follow.
Thank you for this suggestion. We have expanded the Introduction by adding the following paragraph to provide the requested background and better contextualize the detailed sections that follow: Immune checkpoint inhibitors (ICIs) have transformed cancer treatment, yet only a fraction of patients experiences durable benefit, highlighting the need for robust predictive biomarkers that integrate both tumor-intrinsic and host-immune factors. Peripheral blood cell ratios—such as the neutrophil-to-lymphocyte ratio (NLR), lymphocyte-to-monocyte ratio (LMR) and platelet-to-lymphocyte ratio (PLR)—have gained prominence because they reflect the balance between tumor-promoting inflammation and antitumor immune competence, are readily derived from routine complete blood counts, and can be monitored over time to capture changes in systemic immunity. Equally critical are tumor-intrinsic biomarkers such as PD-L1 expression, which directly measures engagement of the PD-1/PD-L1 axis in the tumor microenvironment and remains the most widely implemented assay for selecting patients for anti-PD-1/PD-L1 therapies. In parallel, circulating tumor DNA (ctDNA) provides a minimally invasive, dynamic readout of tumor burden and early treatment effect, with rapid declines in ctDNA levels after ICI initiation correlating with superior progression-free and overall survival independently of PD-L1 status or tumor mutational burden. More exploratory approaches, including rheological parameters that assess tumor-induced alterations in blood flow and viscosity, aim to capture the impact of systemic inflammation on immune-cell trafficking and endothelial interactions. By combining tumor-intrinsic markers, systemic inflammation indices, and real-time circulating measures, a multi-dimensional biomarker framework can more precisely stratify patients and guide personalized immunotherapy strategies.
- Figure 1 contains several incorrect labels or typographical errors that impede its ability to convey helpful information; a redesign or thorough correction is recommended.
Thank you for pointing this out. We have thoroughly corrected all labels and typographical errors and redesigned the figure accordingly. Please note that, after adding a new figure earlier in the manuscript, this is now labeled Figure 2 rather than Figure 1. We have also expanded and clarified the legend beneath Figure 2 to facilitate its interpretation. We appreciate your careful review and believe these changes greatly improve the figure’s clarity.
- Some concepts or roles of biomarkers (e.g., LDH as an indicator of cell damage/tumor burden, or MDSCs' immunosuppressive role) are introduced, then elaborated upon in the "Molecular Mechanisms" section, and sometimes reiterated in the "Discussion." For a "Mini Review," streamlining these explanations by perhaps integrating the primary mechanism when the biomarker is first introduced could enhance conciseness and reduce repetition.
Thank you for this suggestion. In the revised manuscript, we have reorganized our presentation of biomarkers by introducing each marker’s primary mechanism at first mention before proceeding to the more detailed “Molecular Mechanisms” section. We have also consolidated overlapping text in the “Discussion” to avoid restatement and maintained overall brevity by grouping related biomarkers and their functions in a single, clearly labeled introductory paragraph.
- A more appropriate placement of Table 1 and Table 2 could be considered to ensure they are introduced in a context that optimally supports the surrounding text and the overall flow of the review.
Thank you for this suggestion. In the revised manuscript, we have split the original Table 1 into two separate tables, now numbered Table 2 and Table 3, to distinguish preclinical and clinical studies more clearly. We have also added brief introductory paragraphs before each table to provide appropriate context and ensure a smooth transition in the text. We appreciate your feedback, which has improved the clarity and flow of our review.
- The text notes that changes in hemorheological parameters are observed in many diseases. When proposing them as biomarkers for immunotherapy response, a key challenge is dissecting changes specific to the tumor-immunotherapy interaction versus systemic changes due to general inflammation, or cancer progression itself, independent of immunotherapy efficacy. This point could be highlighted more.
Thank you for this insightful comment. In response, we have expanded the relevant section of the manuscript to more clearly emphasize the challenge of distinguishing hemorheological changes that are specific to tumor–immunotherapy interactions from those arising due to general inflammation or cancer progression, independent of treatment efficacy. We now explicitly discuss the multifactorial nature of hemorheological alterations and highlight the need for longitudinal monitoring, integration with established biomarkers, and the use of advanced analytical methods to improve specificity. These additions underscore the complexity of interpreting these parameters and strengthen the rationale for their further investigation as potential immunotherapy-specific biomarkers.
- The discussion on hematocrit value in hematological neoplasms and the mention of COVID-19 vaccinations in multiple myeloma patients are quite specific to hematological contexts. While broadly related to blood properties, direct relevance as biomarkers for immunotherapy response in solid tumors is less clear and could potentially dilute the focus of this section for a mini review.
Thank you for the valid comment regarding the discussion of hematocrit in hematological malignancies and the mention of COVID-19 vaccination in patients with multiple myeloma in relation to biomarkers of immunotherapy response in solid tumors. In the manuscript, we highlight that parameter such as whole blood viscosity (WBV), plasma viscosity, and erythrocyte aggregability are considered potentially promising biomarkers [References 1–3, 51]. In solid tumors, these factors are particularly relevant because they directly influence lymphocyte infiltration and the efficacy of immune checkpoint inhibitors (ICIs) [Reference 49]. For example, in a study on hepatocellular carcinoma (HCC), a solid tumor, higher WBV (>16.0 cP) was associated with poorer overall survival (OS) and progression-free survival (PFS) in patients treated with nivolumab (p=0.069 and p=0.067, respectively) [Reference 52]. The mention of hematocrit in hematological malignancies [Reference 48] may also be useful, as it illustrates a broader principle: the physical properties of blood can modulate immunotherapy effectiveness. This principle is equally applicable to solid tumors, where altered blood viscosity or hematocrit levels may impair microcirculatory flow and hinder immune cell access to tumor sites [Reference 51]. Therefore, rather than detracting from the focus, this discussion underscores the potential of hemorheological markers as universal tools for evaluating immunotherapy response. The reference to COVID-19 vaccination in patients with multiple myeloma, where increased WBV and thrombotic risk were observed [Reference 47], may initially seem specific to hematologic cancers. However, it highlights a key point: systemic inflammatory and rheological changes—induced by disease, treatment, or external factors such as vaccination—can influence blood-based biomarkers relevant to immunotherapy. Similar disturbances are observed in solid tumors; for instance, cancer-induced hyperviscosity has been reported across various malignancies, often compensated by anemia but potentially exacerbated by elevated hematocrit levels [Reference 51]. Such changes can affect immunotherapy through impacts on immune cell dynamics and tumor microenvironment (TME) conditions, such as hypoxia and inflammation, which are known to influence ICI efficacy [References 1–4]. Although the multiple myeloma example originates from a hematological context, it serves as a proof of concept that rheological alterations have systemic significance that deserves exploration in solid tumors, where data remain limited. We have shown that hemorheological changes are not confined to hematologic cancers—for example, patients with newly diagnosed solid tumors often exhibit increased plasma viscosity and red blood cell aggregation [Reference 51]. In HCC, the association between elevated WBV and poorer immunotherapy outcomes [Reference 52] directly links these parameters to solid tumor treatment, even if the evidence is still preliminary. In our view, the inclusion of examples concerning hematocrit in hematological malignancies and the effects of COVID-19 vaccination in multiple myeloma patients does not weaken the focus of our review; rather, it enriches it by demonstrating how hemorheological parameters reflect systemic changes that may influence immunotherapy efficacy in solid tumors. By linking them to mechanisms such as immune cell trafficking and tumor microenvironment oxygenation—as supported by data from HCC [Reference 52] and broader rheological findings [Reference 51]—their discussion remains consistent with the aim of the review, which is to explore low-cost, accessible blood-based biomarkers.
- The review rightly states that "the data for solid tumors affecting rheological blood properties is firmly limited." While the potential is highlighted, the specific mechanisms by which immunotherapy in solid tumors alter these parameters to predict response needs careful exploration.
Thank you for this important observation. In the revised version of the manuscript, we have further addressed the need to explore specific mechanisms by which immunotherapy may influence hemorheological parameters in solid tumors. While current data remain limited, we propose that immunotherapy-induced modulation of the tumor microenvironment—including changes in vascular perfusion, immune cell trafficking, and cytokine-mediated inflammation—may directly or indirectly impact blood viscosity, plasma composition, and erythrocyte behavior. We highlight the study in hepatocellular carcinoma (HCC) as a preliminary example supporting this link, along with associations between hemorheological parameters and inflammatory markers such as NLR and PLR. These relationships suggest a biologically plausible connection that merits further investigation. We agree that larger studies using standardized methods are needed to validate these mechanisms and confirm the clinical relevance of hemorheological biomarkers in predicting immunotherapy outcomes in solid tumors. To reflect this point more clearly, we have also added a sentence beneath the current Table 1 emphasizing the need for further mechanistic exploration and validation in solid tumors.
- While Figure 4 effectively illustrates the diverse array of blood-based markers alongside FDA-approved tumor-based markers, providing a speculative or evidence-based example of how these different types of markers might be integrated into a multi-parametric predictive model would greatly enhance this section.
Thank you for your insightful observation regarding Figure 4. We agree that providing a speculative or evidence-based example of how different types of biomarkers can be integrated into a multiparametric predictive model would enrich this section. To address this, we have included in Discussion the following evidence-based example illustrating the practical application of such integration:
“The use of multiparametric predictive models in cancer treatment, particularly in the context of immunotherapy, is justified for several reasons based on the analysis of the presented text and general trends in oncology. Integrating immunological markers such as CTLA-4 and PD-L1, inflammatory markers (NLR, PLR, LMR), and rheological parameters such as RDW-SD, MCV, PDW, APTT, P-LCR, and MPV enables more accurate prediction of treatment response and identification of patients most likely to benefit from immune checkpoint inhibitor therapy. [...] Multiparametric models, such as the random forest (RF) model, which incorporates a variety of blood parameters—e.g., RDW-SD, MCV, PDW, CD3+CD8+, APTT, P-LCR, calcium, MPV, CD4+/CD8+ ratio, and AST—demonstrate superior predictive performance compared to conventional models.”
This example not only includes the integration of FDA-approved tumor biomarkers (e.g., PD-L1), but also expands the scope to routine blood-based markers that have shown predictive value in machine learning models such as random forest. The proposed approach highlights the potential of combining data from both tumor-derived and peripheral blood biomarkers to improve the accuracy and clinical utility of predictive models in immuno-oncology.
- Thank you for your observation. In response a new subchapter titled " Integrated significance of inflammatory markers and blood rheological parameters in assessing the response to immunotherapy" has been added to clarify the structure and ensure that the content previously included in Section 3 is appropriately organized and contextualized.
Round 2
Reviewer 3 Report
Comments and Suggestions for Authors
Authors addressed my comments in a good manner. The manuscript is suitable for publication
Author Response
Dear Reviewer,
thank you very much for your kind feedback. We sincerely appreciate your time and consideration.
Sincerely,
Reviewer 5 Report
Comments and Suggestions for Authors
Thank you for the authors’ detailed reply and evident determination to improve the manuscript. I appreciate the effort put into addressing the initial comments. However, upon further review, several issues still require attention:
- Regarding the graphical abstract, the organization structure remains unclear, and the function of the visual flow is not apparent.
- Repetitive content has been identified in Figure 1. Please locate and address this within the manuscript.
- Figure 2 (previously Figure 1) still contains incorrect labels or typographical errors that hinder its clarity and ability to convey helpful information effectively.
- The overall structure and logical flow of the manuscript still lack organization, leading to confusion.
Author Response
Dear Reviewer,
Thank you for your valuable feedback. Please find below our point-by-point responses to your comments. We have carefully addressed all the concerns raised and made the appropriate revisions in the manuscript. We hope the changes improve the clarity and quality of our work.
- Regarding the graphical abstract, the organization structure remains unclear, and the function of the visual flow is not apparent.
We have revised the Graphical Abstract to address your concerns. Specifically, we have reorganized the structure by dividing the markers into clear categories and arranging them in a more logical order. Additionally, we simplified the graphic design to enhance readability and improve the visual flow. We believe these changes make the Graphical Abstract clearer and more informative.
2. Repetitive content has been identified in Figure 1. Please locate and address this within the manuscript.
Thank you for pointing this out. We agree that Figure 1 contained repetitive content already described in the manuscript. Therefore, we have decided to remove Figure 1 to avoid redundancy and improve the overall clarity of the text.
3. Figure 2 (previously Figure 1) still contains incorrect labels or typographical errors that hinder its clarity and ability to convey helpful information effectively.
Thank you for your observation. We have carefully revised Figure 2 (Figure 1 in actual version of the manuscript) to correct all labeling issues and typographical errors. Additionally, we improved the overall layout to enhance readability and ensure that the figure conveys the intended information more clearly and effectively.
4. The overall structure and logical flow of the manuscript still lack organization, leading to confusion.
Thank you for your comment. To improve the structure and logical flow of the manuscript, we have revised the titles of several subsections to better reflect their content and to introduce clearer organization regarding the described biomarkers. For example, we renamed Section 3 to “Blood-based biomarkers” and added a new subsection titled “New players in immunotherapy: molecular basis of emerging peripheral blood biomarkers.” We believe these changes help clarify the manuscript’s structure and guide the reader more effectively through the discussed topics.